# Data-Driven Multi-Objective Optimization Approach to Loaded Meshing Transmission Performances for Aerospace Spiral Bevel Gears

**DOI:** 10.3390/ma17051185

**Published:** 2024-03-04

**Authors:** Zhenyu Zhou, Wen Shao, Jinyuan Tang

**Affiliations:** 1State Key Laboratory of Precision Manufacturing for Extreme Service Performance, College of Mechanical and Electrical Engineering, Central South University, Changsha 410083, China; 2AECC Hunan Aviation Powerplant Research Institute, Zhuzhou 412002, China

**Keywords:** aerospace spiral bevel gears, loaded meshing transmission performances, machine tool settings, numerical loaded tooth contact analysis (NLTCA), multi-objective optimization (MOO)

## Abstract

Loaded meshing transmission performance optimization has been an increasingly significant target for the design and manufacturing of aerospace spiral bevel gears with low noise and high strength. An innovative data-driven multi-objective optimization (MOO) method is proposed for the loaded meshing transmission performances of aerospace spiral bevel gears. Data-driven tooth surface modeling is first used to obtain a curvature analysis of loaded contact points. An innovative numerical loaded tooth contact analysis (NLTCA) is applied to develop the data-driven relationships of machine tool settings with respect to loaded meshing transmission performance evaluations. Moreover, the MOO function is solved by using an achievement function approach to accurate machine tool settings output, satisfying the prescribed requirements. Finally, numerical examples are given to verify the proposed methodology. The presented approach can serve as a powerful tool to optimize the loaded meshing transmission performances with higher computational accuracy and efficiency than the conventional methods.

## 1. Introduction

Spiral bevel gears are increasingly demanded for torque or speed transformation in many industrial applications [1]. Especially in the aerospace industry, in order to satisfy the increasingly high transmission requirements of low noise and high strength, loaded meshing transmission performance evaluations are needed to be guaranteed during the design and manufacturing. Actually, loaded meshing transmission performance evaluations have always been required for spiral bevel gears. For instance, no loaded edge contact, no stress concentration, a reasonable loaded transmission error, and a good contact pattern are usually prescribed as important evaluations for noise and strength [2]. In recent research on the tooth contact analysis (TCA) technique, the simulated loaded tooth contact analysis (SLTCA) based on an economical finite element software package has always been an important access to loaded meshing interface state prediction and contact mechanical performance optimization before the actual flank manufacturing [3,4,5]. Here, loaded meshing transmission performance evaluations mainly include contact pressure, contact stress, root bending stress, the loaded contact pattern, and the loaded transmission error [6,7]. Argyris et al. [4] developed a computerized method to analyze the tooth contact and stress of spiral bevel gears.

In the past few decades, in order to acquire high-loaded meshing transmission performances, many researchers have proposed many methods for complex loaded contact behavior analysis [8,9,10,11]. Gleason Works [12] first applied a prescribed parabolic transmission error curve to improve the contact motion path. Litvin [13] developed an accurate geometric tooth flank model of a spiral bevel gear with the applications of the differential geometry method and gearing theory, where the loaded contact stress and root bending stress were determined. Wink and Serpa [14] proposed an accurate composite deformation determination of the whole contact flank. The sum of deformation caused by the load at each point was calculated to determine the loaded transmission error. Wu and Tsai [15] developed loaded contact pressure distribution, which was always one of the important standards for loaded meshing transmission performances. Kolivand and Kahraman [16] proposed a prediction approach to the loaded contact pattern and time-varying meshing stiffness. However, the loaded tooth surface contact problem has not been effectively dealt with since loaded contact stresses underneath the tooth surface have not been taken into account.

More recently, many gear designers have paid attention to loaded meshing transmission performance optimization design considering the actual manufacturing requirements. Machine tool settings have always been used as basic design variables for tooth surface design and loaded meshing performance assessment [13,17]. Simon [18] focused on loaded meshing performances and made improvements by analyzing the influence on the tooth flank from errors. Sugimoto et al. [19] performed analytical and experimental investigations on the loaded transmission error. The actual contact ratio was finally determined by them through numerical comparisons between the theoretical and experimental results. Su et al. [20] proposed a design and analysis approach for spiral bevel gears with a seventh-order transmission error, where the spiral bevel gear could be face-milled on a universal cradle-style or computerized numerical control (CNC) hypoid generator. Undoubtedly, to obtain high-performance evaluations satisfying the complex conditions, the data-driven optimization or modification of the tooth surface played an influential role in integrating the design with the actual manufacturing [21,22]. Stadtfeld and Gaiser [23] developed a universal motion concept (UMC) and ultimate motion graph (UMG), which were capable of correcting the complex tooth surface in multi-axis CNC gear generators. Fan et al. [24,25] developed a tooth surface form error correction method with a high order for face-hobbed or face-milled spiral bevel gears. Fan [26] obtained high-performance tooth surface evaluation using UMC higher-order universal motions. Ding et al. [27] presented a tooth surface optimization method using machine tool settings modification and data-driven correction considering the measured cutter geometric error. Here, it aimed at tooth flank optimization considering both micro-geometry and loaded meshing transmission performance evaluations. Artoni et al. [28,29] constructed a robust least squares model for the modification of the target tooth surface. Formulations for optimizing the contact patterns and loaded transmission errors were provided by employing an ease-off topography correction. Then, a set of accurate machine tool configurations were ascertained employing the Levenberg–Marquardt algorithm. Artoni et al. [30] innovatively devised a flank morphology correction technique with free-form using sensitivity analysis and nonlinear solution methods.

Contemporary investigations within the realm of gear research have predominantly centered on a variety of tooth contact mechanical performance evaluations, such as contact pressure, stress, and loaded contact pattern and transmission error [31,32]. However, some researchers [33,34] have restricted their investigations to singular evaluations of loaded meshing performance in isolation, thus inadvertently precluding a holistic assessment of the multifarious evaluation parameters, only separately considering one of the loaded meshing performance evaluations [35]. Moreover, in recent MOO designs for loaded meshing performances, the establishment of data-driven functional relationships has not been direct and efficient in terms of the final solution [36,37,38]. Ding [21] proposed indirect data-driven relations between performance evaluations and machine tool settings, employing data-driven operation and approximation of SLTCA numerical results. Similarly, Shao and Ding [39,40] also used these approximate functional relations for the MOO machine tool settings modification process. Artoni [38] neglected the detailed establishment of intricate functional interdependencies vis-à-vis the pertinent key machine tool settings.

Particularly within the aerospace domain, where spiral bevel gears are subjected to conditions of high-speed operation, substantial loads, and intricate environments, there exists an ongoing imperative for heightened and reinforced performance in loaded meshing transmissions. However, recently, integrated design considering both of the multiple evaluations has been difficult because of complex data-driven relations. Here, in full consideration of the requirements of aerospace spiral bevel gears under high speed, substantial load, and complex or even extreme weather conditions, this study attempts to develop a high-performance optimization design. In particular, a novel data-driven MOO computation is introduced herein to determine the required loaded meshing transmission performances. In contrast to the conventional “trial-to-error” approach [21], the proposed methodology incorporates an adaptive adjustment of machine tool settings through the implementation of the MOO design framework. Furthermore, within the purview of this data-driven approach, the amalgamated tooth flank design seamlessly interfaces with the practical manufacturing domain through the optimization of initial machine tool settings. Notably, these machine tool settings are not only harnessed to realize a data-driven tooth flank design, but they are also seamlessly integrated into the hypoid generator for the purpose of executing the tangible manufacturing process. Finally, with data-driven relations and a robust MOO solution, data-driven control and decision for collaborative optimization of the required loaded meshing transmission performances are developed. This endeavor undertakes a series of distinct tasks aimed at realizing the aforementioned objective:(i)In contrast to the conventional approach of SLTCA, the current study employs a novel numerical methodology termed Numerical Loaded Tooth Contact Analysis (NLTCA). This innovative technique serves to delineate data-driven correlations between machine tool settings and evaluations of loaded meshing transmission performance. By doing so, NLTCA effectively forges a significant link between the intricacies of flank design and the real-world transmission performance of spiral bevel gears [1,21].(ii)MOO computation of multiple loaded meshing transmission performance evaluations, which mainly include the spatial distribution of loaded contact pressure, the configuration of loaded contact patterns, the elastic deformation characteristics exhibited during contact interactions, and the quantification of loaded transmission error. This computation can significantly improve the precision and efficiency of the complex manufacturing system for spiral bevel gears [38,39].(iii)Data-driven determination of loaded meshing transmission performances is provided in the form of Hertz contact solution. In this context, the machine tool settings emerge as pivotal yet unspecified parameters governing both the design and manufacturing aspects. This integrated approach, which harmoniously aligns tooth flank design with the manufacturing phase, has the potential to expedite the developmental efficiency of aerospace spiral bevel gear products [40].(iv)By employing the proposed MOO framework for machine tool settings modification, a notable endeavor is undertaken to significantly contribute to contemporary collaborative manufacturing paradigms taking into account both geometric and physical performances [40]. It can extend the recent collaborative manufacturing of a case in that higher loaded contact performances were simultaneously controlled and optimized within the qualified scopes for aerospace spiral bevel gears.

## 2. Data-Driven Tooth Surface Modeling

Due to unique flank flexural behaviors [2], the tooth surface is so complicated that explicit expression of the tooth surface equation cannot be obtained. Actually, to obtain a unified and standard tooth surface expression, the establishment of this mathematical model has become a data-driven simulation mirroring the intricacies of the genuine tooth flank manufacturing process [32]. Recently, accurate modeling has been divided into two steps: (1) cutter blade design, and (2) machine kinematics.

### 2.1. Cutter Blade Design

Within the domain of aerospace spiral bevel gear generation, the process involves the utilization of a cutter head, which, as it revolves around the axis, generates the tooth surface by enveloping the spatial trajectory of its curved surface family while interacting with the gear blank through cutting [2,5]. Notably, the present landscape of cutter blade design comprises three fundamental geometric configurations. Figure 1 depicts a data-driven representation of the geometric shape of the cutter blade, wherein the cutter center system is denoted as ***O***_c_(***X***_c_, ***Y***_c_, ***Z***_c_), with ***Z***_c_ signifying the orientation of the cutter axis. In this context, a comprehensive geometric configuration has been chosen for simulation, subsequently partitioned into two distinct segments: (A) the straight-line component, and (B) the circular arc component. The attributes of the cutter blade, encompassing both its position vector and unit normal vector, are outlined as follows:(A)Straight-line component
(1a)$$r_{p}(\mu_{p},\theta_{p})=\left[\begin{array}{c} (r_{c}\pm \mu_{p}\sin\alpha_{c})\cos\theta_{p}\\ (r_{c}\pm \mu_{p}\sin\alpha_{c})\sin\theta_{p}\\ -\mu_{p}\cos\alpha_{c}\\ 1 \end{array}\right]$$
(1b)$$n_{p}(\theta_{p})=\frac{\partial r_{p}}{\partial u_{p}}\times \frac{\partial r_{p}}{\partial \theta_{p}}=\left[\begin{array}{c} \cos\alpha_{c}\cos\theta_{p}\\ \cos\alpha_{c}\sin\theta_{p}\\ \pm \sin\alpha_{c} \end{array}\right]$$(B)Circular arc component
(2)$$\left\{\begin{array}{l} r_{p}(\lambda_{f},\theta_{p})=\left[\begin{array}{c} (r_{c}\pm \rho_{f}(1-\sin\alpha_{c})/\cos\alpha_{c}\mp \rho_{f}\sin\lambda_{f})\cos\theta_{p}\\ (r_{c}\pm \rho_{f}(1-\sin\alpha_{c})/\cos\alpha_{c}\mp \rho_{f}\sin\lambda_{f})\sin\theta_{p}\\ -\rho_{f}(1-\cos\lambda_{f})\\ 1 \end{array}\right]\left(0\leq \lambda_{f}\leq \frac{\pi }{2}-\alpha_{c}\right)\\ r_{c}=r_{u}\pm \frac{P_{w}}{2},n_{p}(\theta_{p})=\frac{\partial r_{p}}{\partial u_{p}}\times \frac{\partial r_{p}}{\partial \theta_{p}}=\left[\begin{array}{c} \cos\lambda_{f}\cos\theta_{p}\\ \cos\lambda_{f}\sin\theta_{p}\\ \pm \sin\lambda_{f} \end{array}\right] \end{array}\right.$$
where *μ_p_* and *θ_p_* represent the Gaussian parameters of the tooth surface [21]; *r_c_* denotes the cutter point radius; *α_c_* stands for the blade pressure angle; *ρ_f_* signifies the edge radius of the cutter head; *λ_f_* corresponds to the angle of the circular arc; *r_u_* denotes the cutter mean radius; and *P_w_* represents the cutter point width. It is worth noting that the upper and lower signs are used to distinguish between the convex and concave sides of the tooth surface for spiral bevel gears [2], respectively.

**Figure 1 materials-17-01185-f001:**
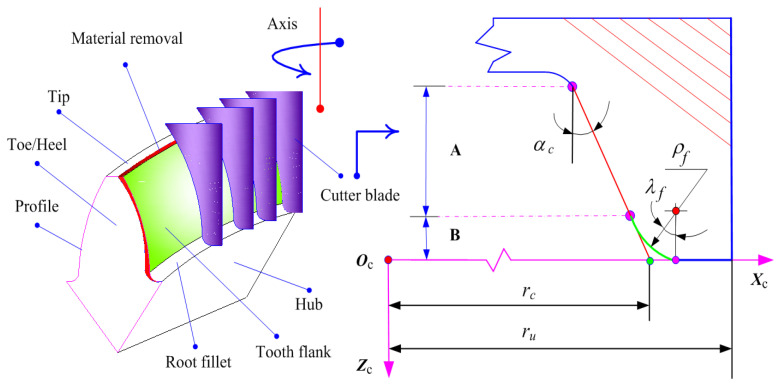
The straight-line blade shape design of the cutter head.

### 2.2. Machine Kinematics

Within the realm of aerospace spiral bevel gear manufacturing, it is possible to conceptualize the spatial trajectory of each cutting blade edge during high-speed rotation as akin to a conical surface. This analogy allows us to envision it as a “tooth” on the generating gear [2]. In light of the kinematic relationship between the simulated generating gear and the workpiece blank, the tooth surface is generated through the enveloping of the surface of the imaged generating gear [2]. The machine kinematics [28,29,30] can be simulated by applying coordinate transformation with respect to the machine tool settings. A basic mathematical model for the tooth surface can be built using a machine kinematics simulation of the cutter blade cutting the work blank, where this process needs to satisfy the gearing theory [33]. In this context, the modeling of the tooth surface is achieved by implementing a kinematic coordinate transformation, transitioning from the tool coordinate system to the gear coordinate system. It is of great significance to determine the coordinate transformation matrices representing the machine kinematics from the cutter blade to the gear work blank.

Figure 2 illustrates the coordinate systems employed to represent the hypoid generator kinematics. These coordinate systems, denoted as *O*_m1_(*x*_m1_, *y*_m1_, *z*_m1_), *O*_a1_(*x*_a1_, *y*_a1_, *z*_a1_), and *O*_b1_(*x*_b1_, *y*_b1_, *z*_b1_) are firmly affixed to the cutting machine center [2]. Furthermore, two mobile coordinate systems, ***O***_1_(*x*_1_, *y*_1_, *z*_1_) and ***O***_c1_(*x*_c1_, *y*_c1_, *z*_c1_), are rigidly connected to the pinion blank and the cradle. These systems rotate about the *z*_b1_ axis and the *z*_m1_ axis, and their rotations are linked to the variable *ϕ*_1_. In addition, the coordinate system *O*_p_(*x*_p_, *y*_p_, *z*_p_) is utilized to illustrate the placement of the head-cutter on the cradle and is associated with the pinion generation process. To achieve this, the coordinate transformation matrix from the cutter coordinate system to the gear coordinate system is calculated:(3)$$\begin{array}{ll} M_{1p} & =M_{1b1}\cdot M_{b1a1}\cdot M_{a1m1}\cdot M_{m1c1}\cdot M_{c1p}\\ & =\left[\begin{array}{cccc} \cos\phi_{1} & \sin\phi_{1} & 0 & 0\\ -\sin\phi_{1} & \cos\phi_{1} & 0 & 0\\ 0 & 0 & 1 & 0\\ 0 & 0 & 0 & 1 \end{array}\right]\cdot \left[\begin{array}{cccc} \sin\gamma_{m1} & 0 & -\cos\gamma_{m1} & 0\\ 0 & 1 & 0 & 0\\ \cos\gamma_{m1} & 0 & \sin\gamma_{m1} & -\Delta X_{D2}\\ 0 & 0 & 0 & 1 \end{array}\right]\cdot \left[\begin{array}{cccc} 1 & 0 & 0 & 0\\ 0 & 1 & 0 & \Delta E_{m1}\\ 0 & 0 & 1 & -\Delta X_{B1}\\ 0 & 0 & 0 & 1 \end{array}\right]\cdot \\ & \left[\begin{array}{cccc} \cos\phi_{c1} & \sin\phi_{c1} & 0 & 0\\ -\sin\phi_{c1} & \cos\phi_{c1} & 0 & 0\\ 0 & 0 & 1 & 0\\ 0 & 0 & 0 & 1 \end{array}\right]\cdot \left[\begin{array}{cccc} 1 & 0 & 0 & S_{r1}\cos q_{1}\\ 0 & 1 & 0 & S_{r1}\sin q_{1}\\ 0 & 0 & 1 & 0\\ 0 & 0 & 0 & 1 \end{array}\right] \end{array}$$
where, *ϕ*_1_ represents the rotational angle of the gear; *γ*_m1_ denotes the machine root angle; X_D2_ corresponds to the increment of the machine center to the rear; Δ*Ε*_m1_ signifies the blank offset; *X*_B1_ refers to the sliding base; *ϕ_c_*_1_ represents the rotational angle of the cradle; *S*_r1_ is the radial setting; *q*_1_ is the basic cradle angle; M_1p_ represents the matrix from the cutter head to the work blank, M_1b1_ represents the matrix from the cradle to the work blank, M_b1a1_ represents the matrix from the machine base to the cradle, M_a1m1_ represents the matrix from the offset position to the machine base, M_m1c1_ represents the matrix from the radial cutter to the machine base, and M_c1p_ represents the matrix from the cutter head to the radial cutter. The sub-matrix of the 4 × 4 matrix ***M***_1p_ is obtained by excluding the final row and column, and it can be represented as:(4)$$\begin{array}{ll} L_{1p} & =L_{1b1}\cdot L_{b1a1}\cdot L_{a1m1}\cdot L_{m1c1}\cdot L_{c1p}\\ & =\left[\begin{array}{ccc} \cos\phi_{1} & \sin\phi & 0\\ -\sin\phi & \cos\phi_{1} & 0\\ 0 & 0 & 1 \end{array}\right]\cdot \left[\begin{array}{ccc} \sin\gamma_{m1} & 0 & -\cos\gamma_{m1}\\ 0 & 1 & 0\\ \cos\gamma_{m1} & 0 & \sin\gamma_{m1} \end{array}\right]\cdot \left[\begin{array}{ccc} 1 & 0 & 0\\ 0 & 1 & 0\\ 0 & 0 & 1 \end{array}\right]\cdot \left[\begin{array}{ccc} \cos\phi_{c1} & \sin\phi_{c1} & 0\\ -\sin\phi_{c1} & \cos\phi_{c1} & 0\\ 0 & 0 & 1 \end{array}\right]\cdot \left[\begin{array}{ccc} 1 & 0 & 0\\ 0 & 1 & 0\\ 0 & 0 & 1 \end{array}\right] \end{array}$$

On the basis of coordinate transformations, a comprehensive numerical simulation of the practical generation process of aerospace spiral bevel gears is conducted. Leveraging position vectors and normal vectors associated with the cutter blade, we adhere to the principles of gearing theory [1] to ascertain the geometry of the tooth flank. The modeling of the tooth flank can be delineated as follows:(5a)$$R_{1}(\mu_{p},\theta_{p},\phi_{c1})=M_{1p}(\phi_{c1})\cdot r_{p}(\mu_{p},\theta_{p})$$
(5b)$$N_{1}(\theta_{p},\phi_{c1})\cdot v^{b-c}=L_{1p}(\phi_{c1})\cdot n_{p}(\theta_{p})\cdot v^{b-c}=0$$
where, ***v****^b−c^* denotes the relative velocity between the cutter and gear blank *b* and cutter *c*. During the gear generating process, the relationship between the angles *ϕ*_1_ and *ϕ*_c1_ is defined as *ϕ*_1_ = m_1c_*ϕ*_c1_, where m_1c_ represents the rolling ratio [2].

## 3. Curvature Analysis of Contact Surface Points

The analysis of the curvature at the contact surface points serves as a fundamental basis for achieving precise TCA solutions [34]. It represents a primary avenue for establishing data-driven functional relationships between machine tool settings and the resulting evaluations of meshing transmission performance [30].

### 3.1. Tooth Contact Point Solution by TCA

In the analytical determination of loaded meshing transmission performance evaluations, TCA is first performed to provide some basis input parameters [34]. To achieve meshing contact positions, the pinion and gear should be rotated by a certain angle to reach the meshing coordinate system [4,5]. The position vector and normal vector in the gear coordinate system are converted into position vector ***R***_m1_(*μ*_p_, *θ*_p_, *ϕ*_c1_) and unit normal vector ***N***_m1_(*θ*_p_, *ϕ*_c1_) in an established meshing coordinate system.

Figure 3 provides a schematic representation of the TCA kinematics for aerospace spiral bevel gears. To accomplish the meshing process, the contact point *P*_2_(*θ*_p_, *ϕ*_c1_) of the pinion tooth surface rotates to a certain angle through the transformation matrix ***M***_2−f_ and the contact point *P*_1_(*θ*_g_, *ϕ*_c2_) of the gear tooth surface rotates to a certain angle by ***M***_1−f_ at the same time. Finally, the pinion and gear make contact at point *P**(*θ*_g_, *ϕ*_c2_, *θ*_p_, *ϕ*_c1_)) in the meshing coordinate systems. The established set of TCA equations must adhere to the following basic conditions:(6a)$$R_{m1}(\mu_{p},\theta_{p},\phi_{c1})={\left(M_{t-f}\right)}_{1}\cdot r_{1}(\mu_{p},\theta_{p},\phi_{c1})$$
(6b)$$N_{m1}(\theta_{p},\phi_{c1})={\left(L_{t-f}\right)}_{1}\cdot n_{1}(\theta_{p})$$

In this context, the complete coordinate transformation matrix from the gear coordinate system to the meshing contact coordinate system is represented as:(7)$$\begin{array}{ll} {\left(M_{t-f}\right)}_{1} & ={\left(M_{t-g}\right)}_{1}\cdot {\left(M_{g-a}\right)}_{1}\cdot {\left(M_{a-b}\right)}_{1}\cdot {\left(M_{b-c}\right)}_{1}\cdot M_{c-f}\\ & =\left[\begin{array}{cccc} 1 & 0 & 0 & -{\left(\Delta l_{X}\right)}_{1}\\ 0 & 1 & 0 & -{\left(\Delta l_{Y}\right)}_{1}\\ 0 & 0 & 1 & -{\left(\Delta l_{Z}\right)}_{1}\\ 0 & 0 & 0 & 1 \end{array}\right]\cdot \left[\begin{array}{cccc} \cos{\left(\Delta {℘}_{Z}\right)}_{1} & -\sin{\left(\Delta {℘}_{Z}\right)}_{1} & 0 & 0\\ \sin{\left(\Delta {℘}_{Z}\right)}_{1} & \cos{\left(\Delta {℘}_{Z}\right)}_{1} & 0 & 0\\ 0 & 0 & 1 & 0\\ 0 & 0 & 0 & 1 \end{array}\right]\cdot \left[\begin{array}{cccc} \cos{\left(\Delta {℘}_{Y}\right)}_{1} & 0 & \sin{\left(\Delta {℘}_{Y}\right)}_{1} & 0\\ 0 & 1 & 0 & 0\\ -\sin{\left(\Delta {℘}_{Y}\right)}_{1} & 0 & \cos{\left(\Delta {℘}_{Y}\right)}_{1} & 0\\ 0 & 0 & 0 & 1 \end{array}\right]\\ & \left[\begin{array}{cccc} 1 & 0 & 0 & 0\\ 0 & \cos{\left(\Delta {℘}_{X}\right)}_{1} & -\sin{\left(\Delta {℘}_{X}\right)}_{1} & 0\\ 0 & \sin{\left(\Delta {℘}_{X}\right)}_{1} & \cos{\left(\Delta {℘}_{X}\right)}_{1} & 0\\ 0 & 0 & 0 & 1 \end{array}\right]\cdot \left[\begin{array}{cccc} 0 & 0 & -1 & 0\\ 0 & 1 & 0 & 0\\ 1 & 0 & 0 & 0\\ 0 & 0 & 0 & 1 \end{array}\right] \end{array}$$

The sub-matrix of the 4 × 4 matrix (***M***_t−f_)_1_ is derived by removing the final row and column of (***M***_t−f_)_1_ and can be represented as:(8)$$\begin{array}{ll} {\left(L_{t-f}\right)}_{1} & ={\left(L_{t-g}\right)}_{1}\cdot {\left(L_{g-a}\right)}_{1}\cdot {\left(L_{a-b}\right)}_{1}\cdot {\left(L_{b-c}\right)}_{1}\cdot L_{c-f}\\ & \left[\begin{array}{ccc} 1 & 0 & 0\\ 0 & 1 & 0\\ 0 & 0 & 1 \end{array}\right]\cdot \left[\begin{array}{ccc} \cos{\left(\Delta {℘}_{Z}\right)}_{1} & -\sin{\left(\Delta {℘}_{Z}\right)}_{1} & 0\\ \sin{\left(\Delta {℘}_{Z}\right)}_{1} & \cos{\left(\Delta {℘}_{Z}\right)}_{1} & 0\\ 0 & 0 & 1 \end{array}\right]\cdot \left[\begin{array}{ccc} \cos{\left(\Delta {℘}_{Y}\right)}_{1} & 0 & \sin{\left(\Delta {℘}_{Y}\right)}_{1}\\ 0 & 1 & 0\\ -\sin{\left(\Delta {℘}_{Y}\right)}_{1} & 0 & \cos{\left(\Delta {℘}_{Y}\right)}_{1} \end{array}\right]\cdot \\ & \left[\begin{array}{ccc} 1 & 0 & 0\\ 0 & \cos{\left(\Delta {℘}_{X}\right)}_{1} & -\sin{\left(\Delta {℘}_{X}\right)}_{1}\\ 0 & \sin{\left(\Delta {℘}_{X}\right)}_{1} & \cos{\left(\Delta {℘}_{X}\right)}_{1} \end{array}\right]\cdot \left[\begin{array}{ccc} 0 & 0 & -1\\ 0 & 1 & 0\\ 1 & 0 & 0 \end{array}\right] \end{array}$$

The rotation matrix ***M***_g−f_ is given by
(9)$$M_{g-f}={\left(M_{g-a}\right)}_{1}\cdot {\left(M_{a-b}\right)}_{1}\cdot {\left(M_{b-c}\right)}_{1}\cdot M_{c-f}$$

In order to achieve the same normal vector, it is necessary to determine the rotational angle (∆℘)_1_=((∆℘_X_)_1_,(∆℘_Ψ_)_1_,(∆℘_Z_)_1_)^Τ^, as well as the translation displacement (∆*l*)_1_=((∆*l*_Ξ_)_1_, (∆*l*_Ψ_)_1_,(∆*l*_Z_)_1_)^Τ^.

To ensure a continuous gear transmission, it is imperative to maintain continuous contact between the tooth surfaces of the pinion and gear [34]. This necessitates that their position vectors and unit normal vectors consistently coincide. In other words, in this contact state, which aligns with the principles of gearing theory, the following relationships hold true:(10a)$$R_{m1}(\mu_{p},\theta_{p},\phi_{c1})=R_{m2}(\mu_{g},\theta_{g},\phi_{c2})$$
(10b)$$N_{m1}(\theta_{p},\phi_{c1})+N_{m2}(\theta_{g},\phi_{c2})=0$$
(10c)$$N_{m1}\cdot v_{12}=0$$
here, the ***v***_12_ denotes the relative velocity of gear with respect to the pinion and can be expressed as
(11)$$v_{12}=v_{1}-v_{2}$$

Six nonlinear scalar equations having six independent parameters such as *μ*_p_, *θ*_p_, *ϕ*_χ1_, *μ*_γ_, *θ*_γ_, and *ϕ*_χ2_ can be identified through the above three vector equations. Finally, by combining the above six nonlinear equations, six unknown variables can be identified by using a nonlinear solver.

The calculation of the driven gear’s rotation angle requires the input of six variables into the TCA equation for accurate computation. The initial step involves the creation of a mathematical model for the tooth surface, based on the provided input data encompassing blank geometry design and machine tool settings. Subsequently, TCA is executed by solving Equation (10) to ascertain whether the contact state aligns with the stipulated design criteria [20]. Finally, TCA evaluations, including the assessment of contact patterns and transmission errors, are derived as part of this analytical process.

### 3.2. Curvature Analysis

In Euclidean space [1], the two surface *Σ*_ι_ (ι=1,2) is represented by ***p***_i_(*ϕ*, *θ*), where (*ϕ*, *θ*) ∈ A represents the fundamental design variable domain. With the modeling of the tooth flank, ***n***_i_ is expressed as
(12)$$n_{i}(\phi ,\theta )=n_{i,\phi }\times n_{i,\theta }\;\mathrm{with}\;n_{i}\neq 0$$

In light of the presumed regularity inherent in the modeling of *Σ*_i_, it can result in:(13)$$E=\frac{d[p_{i}(\phi ,\theta )]}{d\phi }\cdot \frac{d[p_{i}(\phi ,\theta )]}{d\phi },F=\frac{d[p_{i}(\phi ,\theta )]}{d\phi }\cdot \frac{d[p_{i}(\phi ,\theta )]}{d\theta },G=\frac{d[p_{i}(\phi ,\theta )]}{d\theta }\cdot \frac{d[p_{i}(\phi ,\theta )]}{d\theta }$$

The unit normal vector of it can be expressed as
(14)$$n_{i}^{\left[U\right]}(\phi ,\theta )=n_{i}(\phi ,\theta )/\left|n_{i}(\phi ,\theta )\right|=n_{i}(\phi ,\theta )/\sqrt{EG-F^{2}}$$

Then, the first type of basic homogeneous can be represented as
(15)$$\begin{array}{l} L=-\frac{d[n_{i}^{\left[U\right]}(\phi ,\theta )]}{d\phi }\cdot \frac{d[p_{i}(\phi ,\theta )]}{d\phi },M=-\frac{d[n_{i}^{\left[U\right]}(\phi ,\theta )]}{d\phi }\cdot \frac{d[p_{i}(\phi ,\theta )]}{d\phi }=-\frac{d[n_{i}^{\left[U\right]}(\phi ,\theta )]}{d\theta }\cdot \frac{d[p_{i}(\phi ,\theta )]}{d\phi },\\ N=-\frac{d[n_{i}^{\left[U\right]}(\phi ,\theta )]}{d\theta }\cdot \frac{d[p_{i}(\phi ,\theta )]}{d\theta } \end{array}$$

In the course of discretizing and fitting the tooth flank, the tooth flank can be effectively represented as *Σ*_i_ (where *i* = 1,2). Consequently, the definition of *C*_i_, located on *Σ*_i_ (where *i* = 1,2), can be established through the utilization of position vectors [2]
(16)$$C_{i}(s)=p^{\left[C\right]}[\theta^{\left[C\right]}(s),\phi^{\left[C\right]}(s)]$$

With regard to *C*_i_, its unit tangent vector is denoted as
(17)$$\frac{dC_{i}(s)}{ds}=\frac{d\left\{p^{\left[C\right]}[\theta^{\left[C\right]}(s),\phi^{\left[C\right]}(s)\right\}}{d\theta }\cdot \frac{d\left[\theta^{\left[C\right]}(s)\right]}{ds}+\frac{d\left\{p^{\left[C\right]}[\theta^{\left[C\right]}(s),\phi^{\left[C\right]}(s)\right\}}{d\phi }\cdot \frac{d\left[\theta^{\left[C\right]}(s)\right]}{ds}$$

Furthermore, the derivative of the unit normal vector along *C*_i_ is expressed as
(18)$$\frac{dn_{i}^{\left[U\right]}}{ds}=\frac{d[n_{i}^{\left[U\right]}(\phi ,\theta )]}{d\theta }\frac{d\left[\theta^{\left[C\right]}(s)\right]}{ds}+\frac{d[n_{i}^{\left[U\right]}(\phi ,\theta )]}{d\phi }\frac{d\left[\phi^{\left[C\right]}(s)\right]}{ds}$$

In this context, the value of *K*_N_[i] for *Σ*_i_ (where *i* = 1,2) is obtained as
(19)$$K_{N}^{\left[i\right]}=-\frac{dC_{i}(s)}{ds}\cdot \frac{dn_{i}^{\left[U\right]}}{ds}$$

## 4. Determining MOO Relations by NLTCA

Distinguishing the conventional SLTCA [37], the novel NLTCA adopts the widely recognized Hertz contact theory to establish MOO functional relationships between evaluations of the loaded meshing transmission performance and various machine tool settings. Obviously, it is a data-driven design for loaded meshing transmission performances.

### 4.1. Determination of Loaded Contact Ellipse

With tooth flank modeling, the surface curvature can be represented as
(20)$$k_{N}^{\left[i\right]}=\frac{d^{2}r_{i}}{ds^{2}}\cdot n_{f}$$
then, there is ∆s = MP* = ρ, it yields
(21)$$\lambda_{i}=K_{N}^{\left[i\right]}\frac{\Delta s^{2}}{2}=\frac{1}{2}K_{N}\rho^{2}$$

As for the hypoid gear flank Σ_i_ (where i = 1, 2), the normal curvatures K_N_[i], as well as the principle curvatures K_I_[i] and K_II_[i], adhere to a relation that satisfies the Euler equation. The offset, denoted as *λ*_i_ (where i = 1, 2), is ultimately determined as [2]
(22)$$\lambda_{i}=\frac{\rho^{2}}{2}(K_{I}^{\left[i\right]}\cos^{2}q_{i}+K_{II}^{\left[i\right]}\cos^{2}q_{i})(i=1,2)$$
where, *q*_i_ (for i = 1, 2) denotes the angle between MP* and the unit vector ***e***_I_[i] (for i = 1, 2) represents the principle curvature, as illustrated in Figure 4. To define the instantaneous ellipse, a coordinate system (***P****;*τ*, *η*) is established within the tangent plane *Π*. The direction of the vector MP* in the (*τ*, *η*) plane is represented as *θ*_CP_.

Within the boundaries of the loaded tooth contact pattern, wherein the elastic deformation of the loaded tooth flank *w*_ED_ is taken into account, a set of discernible phenomena emerge
(23)$$\lambda_{1}-\lambda_{2}=\pm w_{\mathrm{ED}}$$

In order to ascertain the properties of the instantaneous ellipse, the following relations are taken into accounts
(24)$$q_{1}=\alpha^{\left[1\right]}+\theta_{\mathrm{CP}},q_{2}=\alpha^{\left[2\right]}+\theta_{\mathrm{CP}},\rho^{2}=\tau^{2}+\eta^{2},\cos\theta_{\mathrm{CP}}=\frac{\tau }{\rho },\sin\theta_{\mathrm{CP}}=\frac{\eta }{\rho }$$

Through the application of geometric transformations, the following expression is derived [1]
(25)$$\begin{array}{l} \tau^{2}(K_{I}^{\left[1\right]}\cos^{2}\alpha^{\left[1\right]}+K_{II}^{\left[1\right]}\sin^{2}\alpha^{\left[1\right]}-K_{I}^{\left[2\right]}\cos^{2}\alpha^{\left[2\right]}-K_{II}^{\left[2\right]}\sin^{2}\alpha^{\left[12\right]})+\eta^{2}(K_{I}^{\left[1\right]}\sin^{2}\alpha^{\left[1\right]}+K_{II}^{\left[1\right]}\cos^{2}\alpha^{\left[1\right]}\\ -K_{I}^{\left[2\right]}\sin^{2}\alpha^{\left[2\right]}-K_{II}^{\left[2\right]}\cos^{2}\alpha^{\left[12\right]})-\tau \eta [(K_{I}^{\left[1\right]}-K_{II}^{\left[1\right]})\sin(2\alpha^{\left[1\right]})-(K_{I}^{\left[2\right]}-K_{II}^{\left[2\right]})\sin(2\alpha^{\left[2\right]})]=\pm 2\delta \end{array}$$

The angle α^[1]^, denoting the angle between the coordinate axis *τ* and the unit vector ***e***_I_^[1]^, is subject to arbitrarily selection. For instance, α^[1]^ may be determined by ensuring adherence to the subsequent relation
(26)$$\left.\begin{array}{r} \left(K_{I}^{\left[1\right]}-K_{II}^{\left[1\right]})\sin(2\alpha^{\left[1\right]})-(K_{I}^{\left[2\right]}-K_{II}^{\left[2\right]}\right)\\ \alpha^{\left[2\right]}=\alpha^{\left[1\right]}+\sigma \end{array}\right\}\mapsto \tan2\alpha^{\left[1\right]}=\frac{(K_{I}^{\left[2\right]}-K_{II}^{\left[2\right]})\sin\sigma }{(K_{I}^{\left[1\right]}-K_{II}^{\left[1\right]})-(K_{I}^{\left[2\right]}-K_{II}^{\left[2\right]})\cos\sigma }$$

Equations (25) and (26) illustrate that the projection of the loaded contact pattern onto the tangent plane *Π* manifests itself as an instantaneous ellipse [1], the equation of which can be deduced as
(27)$$B\tau^{2}+A\eta^{2}=\pm w_{\mathrm{ED}}$$

The lengths of major and minor semi-axes of the contact ellipse are denoted as *a*_CP_ and *b*_CP,_ and their expressions are as follows
(28a)$$a_{\mathrm{CP}}=\sqrt{\left|\frac{w_{\mathrm{ED}}}{A}\right|}$$
(28b)$$b_{\mathrm{CP}}=\sqrt{\left|\frac{w_{\mathrm{ED}}}{B}\right|}$$
where,
(29a)$$A=\frac{1}{4}\left(K_{\Sigma }^{\left[1\right]}-K_{\Sigma }^{\left[2\right]}-\sqrt{g_{1}^{2}-2g_{1}g_{2}\cos2\sigma +g_{2}^{2}}\right)$$
(29b)$$B=\frac{1}{4}\left(K_{\Sigma }^{\left[1\right]}-K_{\Sigma }^{\left[2\right]}+\sqrt{g_{1}^{2}-2g_{1}g_{2}\cos2\sigma +g_{2}^{2}}\right)$$
(29c)$$K_{\Sigma }^{\left[i\right]}=K_{I}^{\left[i\right]}+K_{\mathrm{II}}^{\left[i\right]}$$
(29d)$$g_{i}=K_{I}^{\left[i\right]}-K_{\mathrm{II}}^{\left[i\right]}$$

With the determination of the data-driven functional relationship, loaded contact performance evaluations are related to the basic design parameters, namely machine tool settings (*μ*, *θ*, *ϕ*)^[P/G]^, where *P* represents the pinion and *G* represents the gear. It is crucial to emphasize that in the research conducted in [2,21], the machine tool settings were delineated as fundamental design variables, forming the basis for the data-driven optimization of evaluations pertaining to loaded contact performance [34]. Within the framework of MOO focused on improving loaded contact performance, certain constraints [33,34] have been defined. These constraints have been established by drawing upon the overarching prerequisites that are integral to the real-world manufacturing processes of aerospace spiral bevel gears.

Recently, the assessment of the loaded contact pattern has emerged as a crucial evaluation criterion in V-H rolling tests conducted post actual manufacturing, as documented in [28,29]. Generally, the size, direction, and position of the whole loaded contact pattern [30] must conform to specified criteria. It is imperative to avoid loaded edge contact extending beyond the geometric boundary of the entire tooth surface. In light of these stipulations, the loaded contact pattern, denoted as *LCP*, is subject to constraints imposed through innovative boundary conditions, as delineated in the following relationships:(30)$$\begin{array}{c} f_{2}:{\left(\mu ,\theta ,\phi \right)}^{\left[P/G\right]}\mapsto LCP(\mu ,\theta ,\phi )\\ s.t.\;\Omega^{L}\leq \sum_{i}^{N}b_{i}\leq \Omega^{U}\\ {℧}^{L}\leq a_{i-P_{\mathrm{MID}}}\leq {℧}^{U} \end{array}$$

Within the given face width direction, Ω^L^ and Ω^U^ denote the lower and upper boundaries. In the context of the instantaneous contact ellipses, the cumulative sum of the short semi-axis lengths is regarded as a size parameter along the face width direction. In the tooth height direction, the criterion for constraint selection entails opting for the larger semi-axis length associated with the central tooth contact points. This choice is substantiated by its typical representation as the largest dimension among all the instantaneous contact ellipses. The boundaries for these constraints are denoted as ${℧}^{L}$ and ${℧}^{U}$, respectively.

### 4.2. Loaded Contact Pressure Distribution

The assessment of the loaded contact pressure distribution on the tooth surface stands as a primary criterion for gauging the loaded contact strength of the gear [38]. Consequently, its maximum magnitude holds paramount significance in the evaluation of the gear transmission performance, exerting a direct influence on the meshing stability of spiral bevel gears and their fatigue life [4]. To accurately calculate the loaded contact pressure distribution, it is imperative to consider the load distribution in the gear meshing process under various applied conditions.

(1)CASE I: when a single pair of gear teeth engages in loaded tooth contact, the input torque ***M***_INPUT_ is predominantly applied to one specific pair of tooth flanks within the transmission system.

The contact force for aerospace spiral bevel gears can be expressed in terms of the input torque as follows:(31)$$F=\frac{\left\|M_{\mathrm{INPUT}}\right\|}{r_{k}\cos\alpha_{k}\cos\beta_{k}}$$
where
(32a)$$r_{k}=\sqrt{{R_{m1}}^{2}(x)+{R_{m1}}^{2}(y)}$$
(32b)$$\sin\beta_{k}=\frac{1}{2r_{0}}\left[R^{\prime }+\frac{R}{R^{\prime }}(2r_{0}\sin\beta -R)\right]$$
(32c)$$R^{\prime }=\sqrt{{R_{m1}}^{2}(x)+{R_{m1}}^{2}(y)+{R_{m1}}^{2}(z)}$$
(32d)$$R=R_{0}-\frac{B}{2}$$
where, ***M***_INPUT_ signifies the input torque applied to the driven gear; *r*_k_ represents the distance from the contact point *K* to the gear axis; α*_κ_* denotes the gear profile angle of the contact point *K*; *β_k_* represents the spiral angle at the contact point *K*; *r*_0_ is cutter radius; *R*′ corresponds the cone distance of the contact point *K*; *R* stands for the cone distance of the midpoint of the tooth surface; *β* signifies the mean spiral angle; *R*_0_ represents the mean cone distance; B denotes the face width.

(2)CASE II: During the simultaneous engagement of two pairs of gear teeth in a loaded tooth contact process, the input torque ***M***_INPUT_ predominantly applies to two distinct pairs of tooth flanks within the gear transmission system.

It is noteworthy that the total input ***M***_INPUT_ is simultaneously distributed into the gear and pinion flanks, which can be yielded as
(33)$$M_{\mathrm{INPUT}}=M_{1}+M_{2}\Rightarrow F_{1}r_{k1}\cos\alpha_{k1}\cos\beta_{k1}+F_{2}r_{k2}\cos\alpha_{k2}\cos\beta_{k2}=\left\|M_{\mathrm{INPUT}}\right\|$$

In the case of loaded tooth contact, it should be that two sides of a certain gear are matched to the respective sides of the other gear. Thus, as the driving gear undergoes a specific rotational angle, it results in the middle gear having an actual rotation angle that aligns with that of its neighboring tooth flanks. It means that, for the driven middle gear, there exist two of the same unloaded transmission errors. Here, a transmission error is defined as the discrepancy between the actual positions of the output gear [28]. If the pinion and gear are not perfectly rigid and conjugate, the loaded transmission error function is
(34)$$\Delta \phi (\phi_{1})=\phi_{2}(\phi_{1})-\frac{N_{1}}{N_{2}}\phi_{1}$$

It can be transformed as
(35)$$\Delta \phi (F_{1},\phi_{1})-\Delta \phi (F_{2},\phi_{1}+\frac{2\pi }{N_{1}})=0$$
where, *ϕ*_1_ represents the rotation angle of the driving gear; *ϕ*_2_ denotes the rotation angle of the driven gear; *N*_1_ is the number of teeth on the driving gear; *N*_2_ is the number of teeth on the driven gear. Equations (34) and (35) are combined and solved to determine the rotation angle at the critical position where a single pair and two pairs of tooth contact occur, denoted as Φ^[∆]^. Finally, the load distribution throughout the entire gear meshing process is described as
(36)$$F=\left\{\begin{array}{cc} F_{1}, & 0\leq \Delta \phi \leq \Phi^{\left[\Delta \right]}\\ F, & \Phi^{\left[\Delta \right]}<\Delta \phi \leq \frac{2\pi }{N_{1}}-\Phi^{\left[\Delta \right]}\\ F_{2}, & \frac{2\pi }{N_{1}}-\Phi^{\left[\Delta \right]}<\Delta \phi \leq \frac{2\pi }{N_{1}} \end{array}\right.$$

All of the individual instantaneous loaded contact ellipses collectively form the entire contact pattern [34]. This ellipse is defined within a plane coordinate system denoted as *Γ*(*t*, *h*), with its center situated at the determined contact point ***Π***∗(*ϕ*_χ1_, *θ*_π_, *ϕ*_χ2_, *θ*_π_). Here, It is worth noting that this assumption holds, considering that the dimensions *a*_CP_ and *b*_CP_ are relatively small in comparison to the tooth dimensions and the tooth surfaces approach flatness. When considering an ellipse with its center at point ***P****, its axes precisely align with the coordinate axes *h* and *t*. The representation of the contact pressure distribution within the tooth contact pattern can be succinctly expressed as
(37)$$D_{\mathrm{CP}}\left(\eta ,\tau \right)=\left(\begin{array}{ll} \frac{3F}{2\pi a_{\mathrm{CP}}b_{\mathrm{CP}}}\sqrt{1-\frac{t^{2}}{{\left(a_{\mathrm{CP}}\right)}^{2}}-\frac{h^{2}}{{\left(b_{\mathrm{CP}}\right)}^{2}},} & \mathrm{if}\frac{t^{2}}{{\left(a_{\mathrm{CP}}\right)}^{2}}+\frac{h^{2}}{{\left(b_{\mathrm{CP}}\right)}^{2}}\leq 1;\\ 0 & \mathrm{else} \end{array}\right.$$

With the determined functional relationships, loaded meshing transmission performance evaluations are related to basic design parameters, namely machine tool settings (*μ*, *θ*, *ϕ*)^[P/G]^, where *P* represents the pinion and *G* represents the gear. It is worth noting that, in some studies [2,28,29,30], the machine tool settings were set as design variables for the optimization of the loaded contact performance evaluations. Within the framework of this MOO aimed at enhancing loaded contact performance, various constraints have been imposed. These constraints have been formulated with reference to the overarching requirements governing the actual manufacturing of tooth flanks.

When determining the distribution of loaded contact pressure, the primary constraint imposed is mainly on the amplitude, which is limited to:(38)$$\begin{array}{l} f_{1}:{\left(\mu ,\theta ,\phi \right)}^{\left[P/G\right]}\mapsto D_{\mathrm{CP}}(\mu ,\theta ,\phi )\\ s.t.\;D^{L}\leq \left|D_{\mathrm{CP}}\right|\leq D^{U} \end{array}$$
where, *D*^L^ and *D*^U^ represent the lower and upper boundaries of the absolute value on loaded contact pressure |*D*_CP_|, respectively. In this optimization process, the constraint on their loaded contact pressure can be imposed by taking into account both the initial tooth contact points and the overall loaded contact pattern throughout the entire meshing process.

### 4.3. Elastic Contact Deformation

The elastic deformation of the loaded tooth surface provides a direct reflection of the impact of the load on the deflection of the tooth surface in aerospace spiral bevel gears [2,37,38,39,40]. This deformation can lead to changes in both geometric and physical performance, making it imperative to enhance the contact strength and transmission accuracy, particularly in aerospace industrial applications [41,42,43]. Therefore, to reduce the elastic contact deformation [36] as much as possible for excellent meshing performance, it is very necessary to obtain a clear relation between elastic contact deformation and some factors regarding contact ellipse geometry [35]. Ultimately, the elastic contact deformation of the tooth surface is determined based on the contact force *F* and the dimension of the contact ellipse, represented as
(39)$$w_{\mathrm{ED}}=\frac{3F}{2E^{*}\pi a_{\mathrm{CP}}}J_{0}$$
where there exists the following relation expressions [44]
(40)$$\frac{1}{E^{*}}=\frac{1-{u_{1}}^{2}}{E_{1}}+\frac{1-{u_{2}}^{2}}{E_{2}},J_{0}=\int_{0}^{\pi /2}\frac{1}{{\left(\sin^{2}\xi +\lambda^{2}\cos^{2}\xi \right)}^{1/2}}d\xi$$
where, *E** represents the equivalent elastic modulus of the gear material, *ξ* denotes the basic phase angle of the point on the ellipse to the long semi-axis; *E*_1_ and *E*_2_ correspond to the elastic moduli of the pinion and gear materials, respectively; *u*_1_ and *u*_2_ represent the Poisson’s ratios of the pinion and gear materials. *J*_0_ represents the first kind of complete ellipse integral.

In consideration of the elastic contact deformation, by referring to the expression in Equation (39), its amplitude is mainly constrained as
(41)$$\begin{array}{l} f_{3}:{\left(\mu ,\theta ,\phi \right)}^{\left[P/G\right]}\mapsto w_{\mathrm{ED}}(\mu ,\theta ,\phi )\\ s.t.\;{\left(w_{\mathrm{ED}}\right)}^{L}\leq w_{\mathrm{ED}}\leq {\left(w_{\mathrm{ED}}\right)}^{U} \end{array}$$
where, (*w*_ED_)^L^ and (*w*_ED_)^U^ represent the lower and upper boundaries. When the elastic contact deformation falls below the minimum prescribed threshold, it indicates that no contact has taken place. Conversely, if the elastic contact deformation exceeds the maximum prescribed threshold, it implies that the contact deformation is too significant to meet the geometric accuracy requirements of tooth surface manufacturing.

### 4.4. Loaded Transmission Error

Loaded transmission error stands as a well-recognized factor significantly contributing to noise and vibration during the meshing process of spiral bevel gears [2,38]. This heightened concern regarding noise and vibration arises not solely from environmental considerations but is also driven by customer expectations [29,30]. This concern is particularly pronounced in the design of aerospace spiral bevel gears, where complex conditions such as heavy loads, intricate operating requirements, and high rotational speeds are encountered. In such scenarios, tooth contact deformation and load distribution can wield considerable influence over meshing characteristics, leading to additional transmission errors [39,40]. The foremost objective, without a doubt, revolves around the minimization of transmission error, thereby not only achieving diminished noise levels but also heightened strength. To pursue this objective, the transmission error ∆*ϕ*_1_ under no load conditions for aerospace spiral bevel gears is determined, guided by the rotational angles of the driving and driven gears. To comprehensively account for the influence of elastic deformation caused by the applied load on the transmission error, the loaded transmission error must be meticulously computed. This necessitates presenting the loaded transmission error while taking into consideration the elastic deformation occurring at the contact point position.
(42)$${\left(\Delta_{\mathrm{ED}}\right)}^{*}=\frac{w_{\mathrm{ED}}}{r_{k}\cos\alpha_{k}\cos\beta_{k}}$$

Therefore, the comprehensive transmission error is designated as
(43)$$\delta \phi_{1}=\Delta \phi \left(\phi_{1}\right)+{\left(\Delta_{\mathrm{ED}}\right)}^{*}$$

Finally, the comprehensive transmission error can be ascertained through the integration of a collection of data concerning loaded contact deformation and load distribution.

The loaded transmission error, serving as a primary contributor to gear noise and vibration, carries substantial significance in practical gear transmission systems. Here, by referring to Equation (43), there exists the following optimization
(44)$$\begin{array}{l} f_{4}:{\left(\mu ,\theta ,\phi \right)}^{\left[P/G\right]}\mapsto \delta \phi_{1}(\mu ,\theta ,\phi )\\ s.t.\;\delta^{L}\leq \delta \phi_{1}\leq \delta^{U} \end{array}$$
where, *w*^L^ and *w*^U^ represent the lower and upper boundaries, respectively. While the primary aim is typically to minimize the magnitude of the loaded transmission error as much as possible, the lower boundary is also set as *w*^L^ by considering the actual manufacturing requirements.

## 5. Data-Driven MOO of Loaded Meshing Transmission Performances

In recent collaborative performance optimizations [21,38,39], the data-driven relations based on the SLTCA solution could directly affect the efficiency of MOO computation. Using the NLTCA-based Hertz contact method, there are data-driven determinations of the loaded meshing transmission performances relating to machine tool settings. To be different from the conventional separate performance optimization [39,40], this represents a significant and innovative effort in MOO design for aerospace spiral bevel gears, particularly when striving to achieve superior loaded meshing transmission performance. Furthermore, this approach has the potential to markedly enhance the efficiency of the current collaborative optimization processes [22,29]. Moreover, an innovative data-driven MOO is proposed to make the automatic and collaborative optimization of loaded meshing transmission performances a reality for aerospace spiral bevel gears.

### 5.1. MOO Model

To this end, by integrating the optimization of each of the above sub-objectives, the MOO problem on the loaded meshing transmission performances of aerospace spiral bevel gears is represented as
(45)$$\begin{array}{l} f^{\left[\mathrm{MOO}\right]}:=[f_{1},f_{2},f_{3},f_{4}]\mapsto {\left(\mu ,\theta ,\phi \right)}^{\left[\mathrm{MOO}\right]}\\ s.t.\;D^{L}\leq \left|D_{\mathrm{CP}}\right|\leq D^{U}\\ \;\;\;\Omega^{L}\leq \sum_{i}^{N}b_{i}\leq \Omega^{U};{℧}^{L}\leq a_{i-P_{\mathrm{MID}}}\leq {℧}^{U}\\ \;\;\;{\left(w_{\mathrm{ED}}\right)}^{L}\leq w_{\mathrm{ED}}\leq {\left(w_{\mathrm{ED}}\right)}^{U};\delta^{L}\leq \delta \phi_{1}\leq \delta^{U} \end{array}$$

In the collaborative optimization of the multiple performance evaluations, Equation (45) is solved to determine the qualified gear design parameters (*μ*, *θ*, *ϕ*)^[MOO]^ that fulfill the specified conditions. Along with the determined design variables, it can be used to obtain a solid model, which is a basic input for SLTCA. It is notable that, in the unknown design variable (*μ*, *θ*, *ϕ*)^[MOO]^, in consideration of product cost and efficiency, the machine tool settings relating to the basic motion parameter *ϕ* are selected to execute the machine kinematics, and the other parameters (*μ*, *θ*) are generally used to control the cutter kinematics.

### 5.2. MOO Solution

In recent MOO algorithms for spiral bevel gear tooth flank design [44,45], evolutionary methods, particularly those rooted in genetic algorithms, have been commonly employed [38] to ensure the attainment of stable numerical solutions for MOO problems. However, collaborative performance optimization is always achieved at a cost of serious computational burden [21,39]. Here, the achievement function approach, a classical MOO solution [38], was chosen due to its computational efficiency.
(46)$$\underset{M\in F}{\min}(W(\omega ;(\mu ,\theta ,\phi ))=\sum_{j=1}^{4}\omega_{j}f^{\left[\mathrm{MOO}\right]}(\mu ,\theta ,\phi )=\sum_{j=1}^{4}\omega_{j}f_{j}(\mu ,\theta ,\phi )).$$
where ***ω*** = (*ω*_1_, …, ω_k_) is selected with the condition that *ω*_j_ ≥ 0 (for j = 1, …, 4), and their collective sum equals 1. The detailed setup can be referred to in [21,39].

It is presumed that all objective functions and their corresponding values, denoted as *ϕ*_φ_(*μ*, *θ*, *ϕ*) (j = 1, 2,…, 4), associated with the decision vector (*μ*, *θ*, *ϕ*) ∈ ***S***, commonly referred to as the Prato solution, have been normalized to a relative scale through the utilization of a transformation method [41].
(47)$$X_{j}^{\mathrm{REL}}:=f_{j}^{\mathrm{REL}}(\mu ,\theta ,\phi )=\frac{f_{j}(\mu ,\theta ,\phi )-X_{j}^{L}}{X_{j}^{U}-X_{j}^{L}}\times 100\%.$$
where *X* represents the constrained condition, characterized by its lower and upper boundaries, namely X^L^ and X^U^. *S*^[P]^ is applied to minimize the designated achievement function *S* (F → R) as
(48)$$\underset{(\mu ,\theta ,\phi )\in F}{\min}S(f_{j},)\to S^{\left[P\right]}(f_{j},(\mu ,\theta ,\phi );\rho ,\omega )$$

In this context, where *ρ* > 1, the detailed expression of *S*^[P]^ as provided by Wierzbicki [42] leads to the final representation of the MOO problem as.
(49)$$S^{\left[P\right]}(f_{j},(\mu ,\theta ,\phi );\rho ,\omega )=-\sum_{j=1}^{k}(\omega_{j}(f_{j}(\mu ,\theta ,\phi )-f_{j}^{*}){)}^{2}+\rho \sum_{j=1}^{k}\max(0,\omega_{j}(f_{j}(\mu ,\theta ,\phi )-f_{j}^{*}){)}^{2}$$
where, *f*_j_* represents the optimization objective onto the Pareto front. All maximums of the achievement function, as indicated by Equation (49), correspond to the set of solutions denoted as (*μ*, *θ*, *ϕ*)∗ ∈ ***P***_S_. These solutions are identified as Pareto optimal due to their inherent monotonicity with respect to the partial order within the objective domain. Subsequently, the optimization process for the MOO of the aerospace spiral bevel gears’ loaded meshing transmission performance involves the application of an interactive reference point method, as described in Refs. [38,39]. This method employs the achievement function to guide and refine the optimization procedure [46,47,48]. Figure 5 illustrates a basic computation flowchart for the MOO solution using the iterative reference point approach. In recent gear design and manufacturing [49,50,51], the nonlinearity and robustness of the MOO solution and its detailed operation can be referred to in Refs. [38,39].

## 6. Numerical Instances

Today, high-performance aerospace spiral bevel gears in the transmission systems of high-power helicopters are exercised to obtain a collaborative MOO design for the required loaded meshing transmission performance evaluations [50,52,53,54]. Moreover, the recent SLTCA based on economical software packages is used to verify the numerical results [43]. To assess the robustness of the presented methodology, various gear and pinion tooth flanks and their loaded meshing transmission performances are determined by using the proposed MOO method.

### 6.1. MOO Performance Evaluations

In recent industrial applications of aerospace spiral bevel gears, especially aerospace engine reducers, the predominant gear type employed is the zero-degree spiral bevel gear, characterized by a spiral bevel angle typically ranging below 8 degrees [5]. Table 1 describes the fundamental geometric gear blank data in real-world manufacturing. Moreover, they are the basic data for three-dimensional solid modeling. To assess the loaded meshing transmission performance parameters, encompassing elements such as the distribution of loaded contact pressure, the configuration of the loaded contact pattern, elastic contact deformation, and loaded transmission error, an authentic industrial application’s face-milled spiral bevel gear is employed. This specific gear specimen, generated via a cradle-based generator, serves as the validating reference for the proposed MOO problem. To ensure the validity of the designated tooth contact mechanical performance parameters, numerical results are obtained using the SLTCA approach via finite element analysis. Subsequently, a comprehensive comparison and analysis of the obtained results are performed [54,55]. Here, with the given basic tooth surface geometric design parameters, data-driven tooth surface modeling is performed. Then, the MOO solution is performed and the performance evaluations from the above two design schemes are given.

With the given basic design data on the gear blank, head-cutter, and hypoid generator, data-driven tooth flank modeling is performed. Utilizing the provided foundational design data pertaining to the gear blank, head-cutter, and hypoid generator, a data-driven approach is employed for tooth flank modeling. As depicted in Figure 6, this modeling encompasses a three-dimensional solid model created through computer-aided design (CAD) software and a finite element model established using the ABAQUS software [43]. To ensure precision in our calculations, the tooth profiles are discretized with solid elements, employing the hexahedral reduced integral element C3D8R. The material properties within the finite element model are configured in accordance with the prescribed material parameters [44], which stipulate a Young’s modulus of 2.09 × 10^5^ MPa and a Poisson’s ratio of 0.3. In the context of the finite element model for aerospace spiral bevel gears, the total count of finite elements encompasses 360,656, with 430,576 nodes. The applied torque is 250 N-M for the gear axis. To optimize computational efficiency, the boundary conditions dictate that the shell bottom is clamped along the circular edge, while the tooth tip, concave, convex, heel, and toe regions remain unrestricted. For comprehensive insights into the design data and procedures, readers are encouraged to refer to the detailed specifications presented in Ref. [43].

Figure 7a illustrates the distribution of tooth contact pressures at loaded contact points within an aerospace spiral bevel gear transmission system. This presentation facilitates a comparative analysis between the outcomes obtained through the proposed numerical approach and the results generated using conventional simulation methods. In Figure 8, the elastic contact deformation is observed on the tooth surface at distinct contact points within the assembly. Specifically, the analysis encompasses the examination of 27 loaded tooth contact points, comprising the selection of the central point of each instantaneous contact ellipse. Subsequently, the elastic contact deformations at these selected points are individually computed and presented for evaluation. In the numerical operation of the SLTCA results, elastic deformation amounts of the whole nodes in instantaneously loaded contact ellipse are respectively extracted and their mean value is considered as the referenced one. Then, SLTCA results are compared with the sampled points from the proposed method. Referring to the time-varying meshing trend in Figure 7b, they can show a similar result. In the proposed numerical approach, it is noteworthy that the maximum observed elastic contact deformation measures 0.060847 mm at the 27th sampled contact point and the minimum is 0.058913 mm. However, in the comparison results, the 16th and 17th contact points have the biggest differences in that the former is less, 0.6571%, and the latter is larger, 0.5427%, than the simulated numerical results, and the other points are basically the same.

Figure 8 shows the loaded contact pattern observed on the tooth surface. In the proposed method in Figure 8a, loaded tooth contact points are first computed and the respective instantaneous contact ellipses are determined. The culmination of this process involves utilizing these instantaneous contact ellipses to construct the loaded contact pattern, conforming to the specified geometric boundaries and the desired scope. The comprehensive loaded contact pattern is primarily situated within the central region of the tooth surface and exhibits an elongated strip-like distribution. This configuration effectively conveys that both the size and orientation of the loaded contact pattern align with the prescribed gear design criteria, encompassing considerations related to noise mitigation and structural strength [28,29,30]. Notably, the presence of loaded edge contact phenomena is conspicuously absent within this pattern. Moreover, by comparisons with the loaded contact pattern from the SLTCA method in Figure 8b, it is obvious that, not only in terms of size but also the direction of the loaded contact pattern, most of them are similar. Here, in the given SLTCA method, the process of determining the loaded contact pattern for the entire tooth contact sequence is notably intricate and the details can be referred to in Ref. [21].

Figure 9 shows the loaded transmission error within the aerospace spiral bevel gear transmission system. In Figure 9a, the numerical method effectively captures the loaded transmission error dynamics over three complete meshing cycles. It is observed that the temporal evolution of this error remains generally stable, albeit with a notable point of volatility encountered during the mid-phase of each meshing cycle. The maximum recorded loaded transmission error stands at 51.246 *μ*rad, under an applied input torque of 250 Nm, while the minimum value registers at 26.473 *μ*rad. Consequently, the amplitude of the entire loaded transmission error fluctuation amounts to 26.473 *μ*rad. This amplitude signifies a favorable vibration performance, aligning with the requisite noise and structural integrity criteria governing gear transmission systems. Furthermore, when a specific meshing cycle is selected, as illustrated in Figure 9b, a comparative analysis is presented between the outcomes derived from the proposed numerical approach and simulations. The results demonstrate a substantial degree of consistency in both trend and amplitude, with the exception of a conspicuous disparity at a particular juncture during the mid-stage of the meshing cycle. Notably, the amplitude discrepancy, as provided by the numerical method, amounts to a reduction of 4.076 *μ*rad when contrasted with the simulation results. Additionally, it is worth highlighting that the discrepancies observed in other loaded tooth contact points remain within the margin of 2.53%.

### 6.2. MOO Output Result

The gear tooth surfaces are concurrently generated employing identical machine tool settings, courtesy of the integrated approach afforded by the proposed MOO method. However, it is important to note that the convex and concave surfaces of the pinion tooth are separately manufactured, each undergoing the face-milling process with distinct machine tool settings. In the case of tooth flank grinding, the gear flank’s concave and convex aspects are simultaneously addressed through a designated set of machine tool settings, while the pinion’s concave and convex flanks are each treated with two distinct sets of machine tool configurations, as elaborated in references [56,57]. Crucially, the proposed methodology constitutes an adaptive, data-driven optimization procedure, driven by a collaborative consideration of the mechanical performance evaluations concerning loaded tooth surface contact [40]. It is pertinent to underscore that this approach, to a considerable extent, underscores an intelligent control paradigm by optimizing the pivotal machine tool settings involved in the manufacturing process.

Within the context of data-driven MOO for enhancing the loaded meshing transmission performances of aerospace spiral bevel gears, Figure 10a,b provides an MOO Pareto front solution for gear machine tool settings. It can show good convergence, iterative quality, and computational efficiency. Moreover, before and after the MOO solution, the corresponding evaluations are significantly improved. The gear MOO output machine tool settings based on a Gleason hypoid generator [56,57] are represented in Table 2. Figure 10c–d showcases the MOO Pareto front solutions for pinion machine tool settings, taking into account the mechanical performance evaluations related to loaded tooth surface contact. The computational results show that the MOO solution, not only for the pinion concave side, but also for the pinion convex side, can attain fast convergence, high iterative quality, and good computational efficiency. Table 3 comprehensively enumerates the MOO-derived machine tool settings for the pinion component, as furnished by the proposed MOO methodology. In conclusion, armed with the determined machine tool settings denoted as (*μ*, *θ*, *ϕ*^)[MOO]^, these settings can be seamlessly translated into actionable directives for the numerical control machine tools, thereby facilitating the precise manufacturing of aerospace spiral bevel gears. This manufacturing process yields gears that unequivocally meet the stringent requirements stipulated for loaded meshing transmission performances.

### 6.3. Numerical Verification

In the literature regarding MOO for enhancing the geometric and physical performances of spiral bevel gear tooth flanks [44,45], there predominantly exists three prevailing algorithmic approaches: (i) the achievement function method [38]; (ii) the nonlinear interval number method [39]; (iii) the Kriging method [11]. Here, the proposed iterative reference point approach is used to achieve a numerical comparison with the above main methods. Table 4 shows the computation efficiency comparison with the recent main MOO algorithms. In terms of the iteration number, the achievement function is at least 1 and 2 less iterations than the proposed method; the largest is 248 for the Kriging method [11]. In terms of iteration time, the proposed iterative reference point method can attain the fastest speed. For the iteration convergence accuracy assessment, the proposed method is still the best. Thus, the computation efficiency evaluations, including iteration number, iteration time, and convergence accuracy, show that the most optimal is the proposed method in this work.

Focusing on the given loaded meshing transmission performance evaluations, especially for their maximum values, Figure 11 shows the numerical comparisons with the recent main MOO algorithms. First, the sub-objective evaluation results of various MOO methods do not have many differences and they can verify the optimization accuracy of the proposed method. Where, the maximum *δ*_max_ is 64.838 μm from the nonlinear interval number method, while the minimum is 56.786 um from the Kriging method [11]. The maximum *LTE*_max_ is 57.216 *μ*rad from the nonlinear interval number method, while the minimum is 51.246 *μ*rad from the proposed method. The maximum (*D*_CP_)_max_ is 1209.844 MPa from the nonlinear interval number method, while the minimum is 1183.225 MPa from the proposed method. It shows that the most optimal MOO design for the spiral bevel gear transmission is the one under consideration.

## 7. Conclusions

In the scope of this research, a data-driven MOO approach, emphasizing the optimization of loaded meshing transmission performance, is proposed for aerospace spiral bevel gears. This optimization is achieved through a systematic adjustment of machine tool settings. The principal findings of this study can be succinctly summarized as follows:

(i)Distinguished from the traditional SLTCA method using economical finite element software [38], the proposed NLTCA offers a reliable and time-efficient avenue for optimizing the performance of loaded meshing transmissions. Furthermore, this numerical approach introduces opportunities for collaborative optimization, encompassing considerations of both geometric and physical performance attributes.(ii)A data-driven, accurate model of the loaded meshing transmission performance MOO in collaborative consideration of the loaded contact pressure distribution, contact elastic deformation, loaded contact pattern, and loaded transmission error is provided. Its inherent versatility empowers gear designers to practically apply these findings in future advanced gear designs by specifying appropriate objective functions. It is noteworthy that the optimization process extends its purview to encompass the tooth flank’s contact fatigue performance, inclusive of factors such as residual stress [57,58], microsurface topography [59], and surface roughness [60]. This approach represents a vital step toward achieving high-performance tooth flank manufacturing for aerospace spiral bevel gears.(iii)The proposed methodology lays a foundational framework for future high-performance design considerations, accounting for complex operational conditions such as high-speed operation, the intricate coupling effects of multiple fields, and lubrication dynamics. It demonstrates the potential to optimize loaded meshing transmission performance within predetermined parameters by fine-tuning machine tool settings. In addition to accounting for various manufacturing errors, the validation of MOO results through enhancements to the employed algorithm [61,62] emerges as a primary focus for future research endeavors [63,64].

## Figures and Tables

**Figure 2 materials-17-01185-f002:**
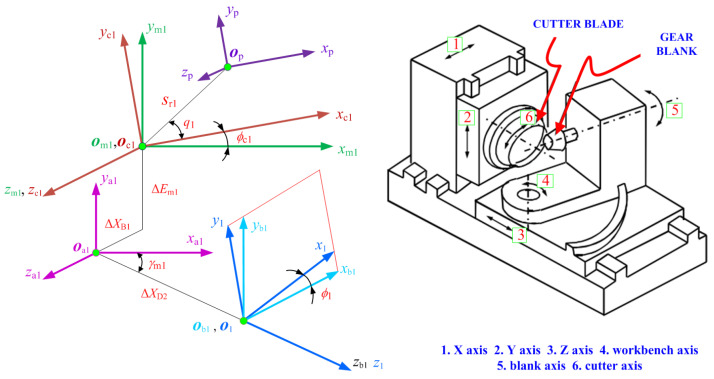
Geometric coordinate system for aerospace spiral bevel gears manufacturing.

**Figure 3 materials-17-01185-f003:**
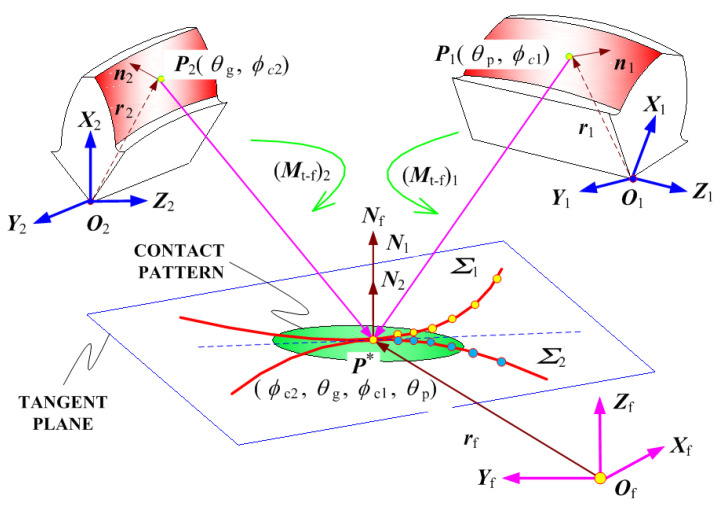
The schematic diagram of TCA kinematics of aerospace spiral bevel gears.

**Figure 4 materials-17-01185-f004:**
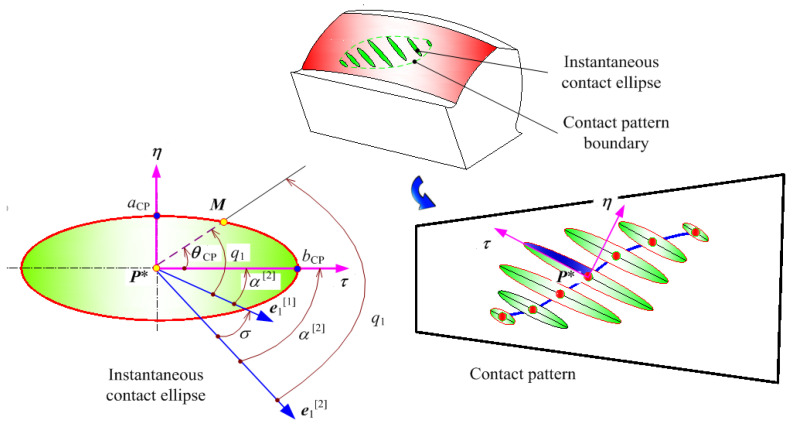
The loaded tooth contact pattern of aerospace spiral bevel gears.

**Figure 5 materials-17-01185-f005:**
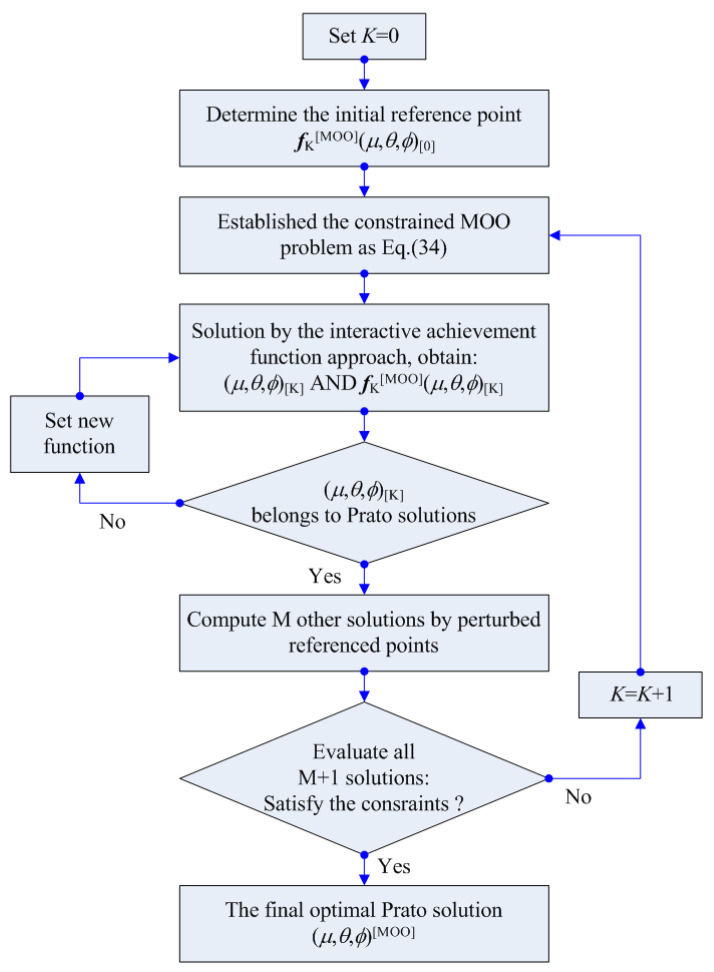
A basic flowchart for the MOO solution using the iterative reference point approach.

**Figure 6 materials-17-01185-f006:**
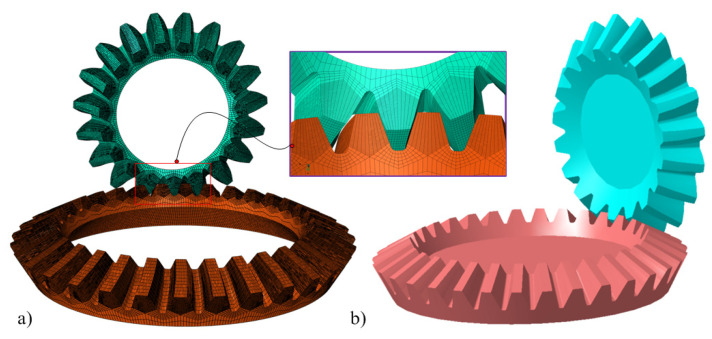
Spiral bevel gear model: (**a**) finite element model; (**b**) 3D solid model.

**Figure 7 materials-17-01185-f007:**
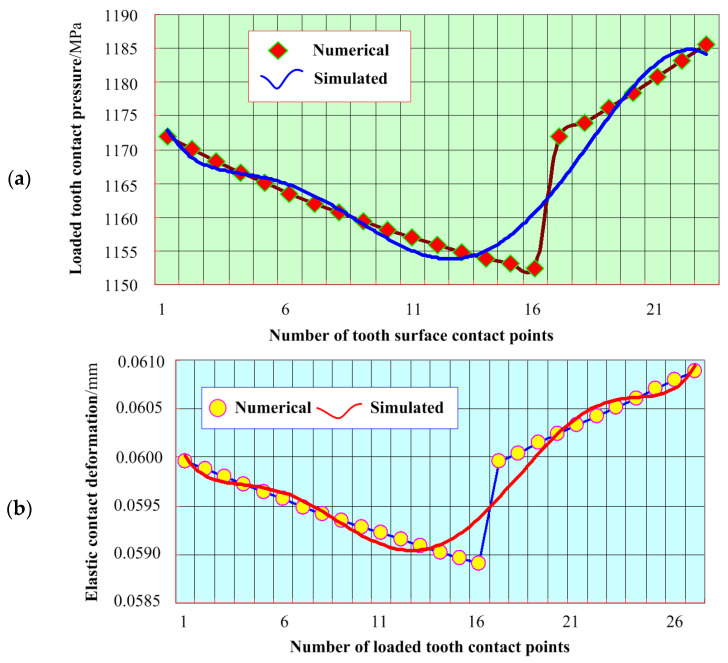
Comparison of MOO results for aerospace spiral bevel gears: (**a**) Tooth contact pressure distribution at contact points; (**b**) The elastic deformation at the contact point of aerospace spiral bevel gear.

**Figure 8 materials-17-01185-f008:**
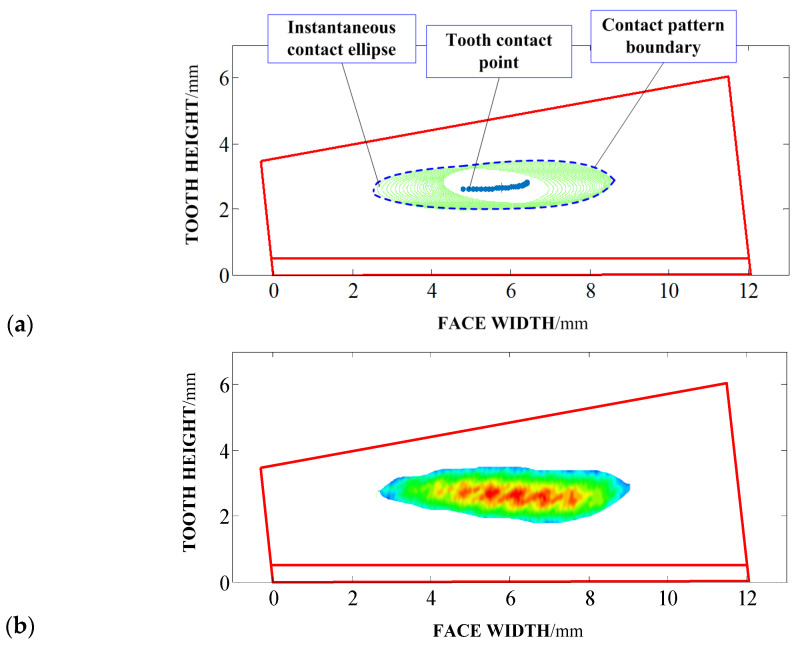
The contact pattern of the tooth surface: (**a**) proposed analytical method, (**b**) SLTCA method.

**Figure 9 materials-17-01185-f009:**
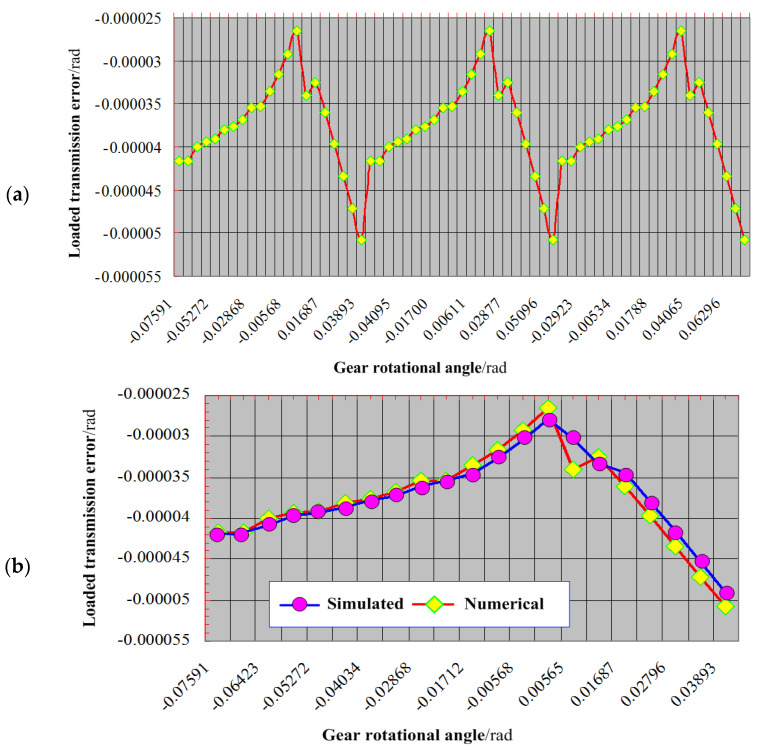
The loaded transmission error: (**a**) proposed analytical method in three meshing cycles, (**b**) result in one cycle by comparison with SLTCA method.

**Figure 10 materials-17-01185-f010:**
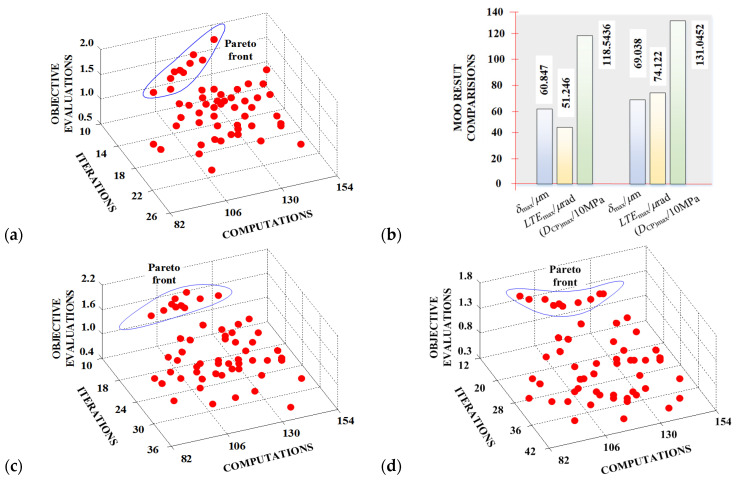
MOO result comparisons: (**a**) Pareto front solution; (**b**) numerical result comparisons before and after MOO; (**c**) MOO Pareto front solution for machine tool settings considering loaded meshing transmission performances for pinion concave side; (**d**) MOO Pareto front solution for machine tool settings considering loaded meshing transmission performances for pinion convex side.

**Figure 11 materials-17-01185-f011:**
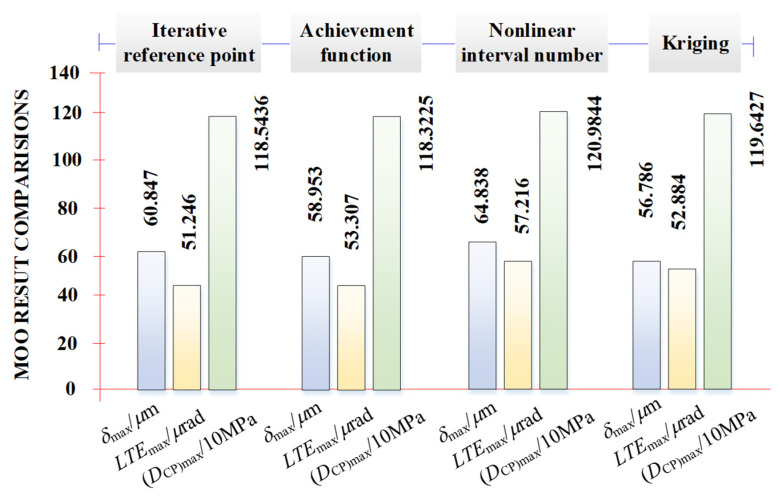
Comparison of the given loaded meshing transmission performance with the recent main MOO algorithms [11,38,39].

**Table 1 materials-17-01185-t001:** Gear blank geometric design parameters of aerospace spiral bevel gears.

Design Parameters	Pinion	Gear
Number of teeth	19	32
Mean normal module (mm)	3.15	3.15
Face width (mm)	20	20
Pressure angle (deg)	21.5	20.5
Root angle (deg)	38.48	47.23
Pitch angle (deg)	40.31	49.29
Face angle (deg)	42.37	51.12
Spiral angle (deg)	7	7
Hand of spiral	LH	RH
Addendum (mm)	3.37	2.63
Dedendum (mm)	3.24	3.99

**Table 2 materials-17-01185-t002:** Gear MOO output machine tool settings.

Machine Tool Settings	Pinion
Cutter diameter (mm)	152.4
Outer tool profile angle (deg)	19
Inside tool profile angle (deg)	21
Cutter point width (mm)	2
Root fillet radius (mm)	0.76
Machine root angle (deg)	47.23
Machine center to back (mm)	0.0058
Sliding base (mm)	0
Blank offset (mm)	0.0045
Radial distance (mm)	119.56625
Velocity ratio	1.3145
Basic cradle angle	40.2541

**Table 3 materials-17-01185-t003:** Pinion MOO output machine tool settings.

Machine Tool Settings	Concave Side	Convex Side
Cutter diameter (mm)	168.07	140.05
Tool profile angle (deg)	19	21
Root fillet radius (mm)	0.54	0.54
Machine root angle (deg)	38.48	38.48
Machine center to back (mm)	0.52	0.43
Sliding base (mm)	−0.32	−0.27
Blank offset (mm)	−0.36	0.15
Radial distance (mm)	124.73455	116.13233
Basic cradle angle (deg)	43.97462	37.69636
Velocity ratio	1.56117	1.52784

**Table 4 materials-17-01185-t004:** Computation efficiency comparison with the recent main MOO algorithms.

	Iteration Number	Iteration Time	Convergence Accuracy
Iterative reference point	84	1.35678 s	6.542 × 10^−9^
Achievement function [38]	82	1.35681 s	2.367 × 10^−8^
Nonlinear interval number [39]	135	12.5437 s	8.645 × 10^−8^
Kriging [11]	248	8.6869 s	9.686 × 10^−9^

## Data Availability

All data that support the findings of this study are included within the article.

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
