# Peer review of "Data-Driven Multi-Objective Optimization Approach to Loaded Meshing Transmission Performances for Aerospace Spiral Bevel Gears"

_materials, 2024, doi:10.3390/ma17051185_

Round 1
Reviewer 1 Report
The article present data-driven approach to optimize the profile of some spiral gears used in aerospace.
There is no reference on materials used in the process and the study itself is clearly oriented toward the fabrication process through 5 axis CNC machines, so this article seems a little out of place in this journal.
If it is done intentionally to get a braoder audience, I recommend to improve the description of the problem and to clarify some points in a more pedantic way than it would have been in a more specialized journal.
In practice, you should define the main terms that you are describing here.
For instance, there is no clear definition of what is rp or any argument to explain why this object named "position vector" should have 4 components. In equation 3, it is presented a set of transformations of coordinate systems in the form of a product of 4x4 matrices, then line 197 it is precised that "the" sub-matrix is obtained by elimination of the last row and the last column. Going to the given reference [21], the vectors are usual 3D vectors, so I do think the presentation here is far to confusing to allow someone not completely involved to follow the computations.
The following part of description of the contact analysis is even more difficult to follow, with a lot of undefined symbols.
I consider that the clear definition of the mathematical symbols used here should be summarized somewhere, with some references for their definition.
You also probably have to clearly refer and expose the well-known facts that you reproduce here for clarity (and you need to explain the details and definitions), and then clearly show where is the originality of your approach, again to a less specialized audience.
Less essential point: you should precise the meaning of "numerical" and "simulated" in all the comparison figures, because the term is somehow confusing.
I don't recommend the publication in materials in the present form.
Author Response
In this revision, we turn our attention to some errors about the grammar and the punctuation, especially about the scientific rigor and the logical thinking. To achieve an excellent quality of the presentation, we have made corresponding corrections. Some cohesion between paragraphs, number right alignment and punctuation of the formula, the clear expression of the problem in the introduction, logical relationship before and after the statement, and unified text format in the whole figures and some other mistakes have been modified, respectively.
Author Response
In this revision, we turn our attention to some errors about the grammar and the punctuation, especially about the scientific rigor and the logical thinking. To achieve an excellent quality of the presentation, we have made corresponding corrections. Some cohesion between paragraphs, number right alignment and punctuation of the formula, the clear expression of the problem in the introduction, logical relationship before and after the statement, and unified text format in the whole figures and some other mistakes have been corrected.
Reviewer 3 Report
This paper “Data-driven multi-objective optimization approach to loaded meshing transmission performances for aerospace spiral bevel gears”, aims to propose a data-driven multi-objective optimization of loaded meshing transmission performances for aerospace spiral bevel gears. Data-driven tooth surface modelling is firstly used to get curvature analysis of loaded contact points. To distinguish with the conventional simulated loaded tooth contact analysis using economical software package, an innovative numerical loaded tooth contact analysis is applied to develop the data-driven relationships of machine tool settings with respect to loaded meshing transmission performance evaluations.
The topic is justified. The paper could be further improved if the following remarks are taken into consideration:
1. ABSTRACT: The text should include numerical results achieved, and a comparison with other methods (if possible).
2. Few grammatical mistakes found in the draft of the article.
3. Introduction section lacks a proper introduction of the whole of the conducted research, background, justification of the research, and major contributions of the study. The contribution may be key fold in the introduction section.
4. Overall, the proposed data-driven multi-objective optimization of loaded meshing transmission methods seems too computational complex, w.r.t time.
5. Results and discussion section seems ok.
6. The motivation is not clear.
7. Discuss the limitations of the proposed method.
8. The conclusion section needs to be redrawn.
minor grammar corrections required
Author Response
- ABSTRACT: The text should include numerical results achieved, and a comparison with other methods (if possible).
RES: We provide a new abstract, as follows:
“In order to meet the low noise and high strength demands from aerospace spiral bevel gears, loaded meshing transmission performance optimization has been an increasingly significant target in recent design and manufacturing. An innovative data-driven multi-objective optimization (MOO) of loaded meshing transmission performances is proposed for aerospace spiral bevel gears. Data-driven tooth surface modeling is firstly used to get curvature analysis of loaded contact points. To distinguish with the conventional simulated loaded tooth contact analysis (SLTCA) using economical software package, an innovative numerical loaded tooth contact analysis (NLTCA) is applied to develop the data-driven relationships of machine tool settings with respect to loaded meshing transmission performance evaluations. It is a bridge between the recent flank design and the actual performance optimization. Moreover, MOO function is solved by using achievement function approach to accurate machine tool settings output satisfying to the prescribed requirements. Finally, the given numerical instance can verify that the proposed methodology can get a higher computational accuracy and efficiency.”
- Few grammatical mistakes found in the draft of the article.
Res: In this revision, we turn our attention to some errors about the grammar and the punctuation, especially about the scientific rigor and the logical thinking. To achieve an excellent quality of the presentation, we have made corresponding corrections. Some cohesion between paragraphs, number right alignment and punctuation of the formula, the clear expression of the problem in the introduction, logical relationship before and after the statement, and unified text format in the whole figures and some other mistakes have been modified, respectively.
- Introduction section lacks a proper introduction of the whole of the conducted research, background, justification of the research, and major contributions of the study. The contribution may be key fold in the introduction section.
Res: In this introduction part, focusing on the contribution of this paper, we provide a following modification:
“In particular, in aerospace application of spiral bevel gears under high-speed, heavy load and complex environment, it has continuously been required the higher and stronger loaded meshing transmission performances. However recently, the integrated design considering both the multiple evaluations has been very difficult because of complex data-driven relations. Here, in full consideration of requirements form aerospace spiral bevel gears under high speed, heavy load and complex or even extreme weather conditions, this study attempts to develop high-performance optimization design. In particular, a new data-driven MOO computation is proposed to determine the required loaded meshing transmission performances. Where, to distinguish with the traditional “trial-to-error” method [21], an adaptive modification of machine tool settings is performed by MOO design. Moreover, in this data-driven operation, the integrated tooth flank design is integrated with the actual manufacturing by optimizing the initial machine tool settings. Where, machine tool settings are used to get a data-driven tooth flank design but also submitted into hypoid generator for executing the actual manufacturing process. Finally, with data-driven relations and robust MOO solution, data-driven control and decision for collaborative optimization of the required loaded meshing transmission performances are developed. This work performs the following specific tasks to achieve this objective:
- i) To distinguish with the conventional SLTCA, new numerical loaded tooth contact analysis (NLTCA) is performed to establish data-driven relationships between machine tool settings and loaded meshing transmission performance evaluations. It can establish an important bridge between the flank design and the actual transmission performances for spiral bevel gears [1,21].
- ii) MOO computation of multiple loaded meshing transmission performance evaluations, which mainly include loaded contact pressure distribution, loaded contact pattern, elastic contact deformation and loaded transmission error. This computation can significantly improve accuracy and efficiency of the complex manufacturing system for spiral bevel gears [38,39].
iii) Data-driven determination of loaded meshing transmission performances is provided in form of Hertz contact solution. Where, the machine tool settings are the basic unknown variables for both design and manufacturing. This method integrating tooth flank design with manufacturing can accelerate the development efficiency of aerospace spiral bevel gear product [40].
- iv) With application of the proposed MOO machine tool settings modification, it can get a significant attempt to the recent collaborative manufacturing considering both geometric and physical performances [40]. It can extend the recent collaborative manufacturing for a case that higher loaded contact performances were simultaneously controlled and optimized within the qualified scopes for aerospace spiral bevel gears.”
- Overall, the proposed data-driven multi-objective optimization of loaded meshing transmission methods seems too computational complex, w.r.t time.
RES: Recently, for complex tooth flank of spiral bevel gears having a special flexural behaviors, the employed MOO algorithm is a very suitable method while combing with accuracy and efficiency. MOO computation of multiple loaded meshing transmission performance evaluations, which mainly include loaded contact pressure distribution, loaded contact pattern, elastic contact deformation and loaded transmission error. This computation can significantly improve accuracy and efficiency of the complex manufacturing system for spiral bevel gears [38,39].
- Results and discussion section seems ok.
RES: Thanks very much!
- The motivation is not clear.
RES: In this revision, focusing on the new contribution, we provided the new explanations, as follows:
“In particular, in aerospace application of spiral bevel gears under high-speed, heavy load and complex environment, it has continuously been required the higher and stronger loaded meshing transmission performances. However recently, the integrated design considering both the multiple evaluations has been very difficult because of complex data-driven relations. Here, in full consideration of requirements form aerospace spiral bevel gears under high speed, heavy load and complex or even extreme weather conditions, this study attempts to develop high-performance optimization design. In particular, a new data-driven MOO computation is proposed to determine the required loaded meshing transmission performances. Where, to distinguish with the traditional “trial-to-error” method [21], an adaptive modification of machine tool settings is performed by MOO design. Moreover, in this data-driven operation, the integrated tooth flank design is integrated with the actual manufacturing by optimizing the initial machine tool settings. Where, machine tool settings are used to get a data-driven tooth flank design but also submitted into hypoid generator for executing the actual manufacturing process. Finally, with data-driven relations and robust MOO solution, data-driven control and decision for collaborative optimization of the required loaded meshing transmission performances are developed. This work performs the following specific tasks to achieve this objective:
- i) To distinguish with the conventional SLTCA, new numerical loaded tooth contact analysis (NLTCA) is performed to establish data-driven relationships between machine tool settings and loaded meshing transmission performance evaluations. It can establish an important bridge between the flank design and the actual transmission performances for spiral bevel gears [1,21].
- ii) MOO computation of multiple loaded meshing transmission performance evaluations, which mainly include loaded contact pressure distribution, loaded contact pattern, elastic contact deformation and loaded transmission error. This computation can significantly improve accuracy and efficiency of the complex manufacturing system for spiral bevel gears [38,39].
iii) Data-driven determination of loaded meshing transmission performances is provided in form of Hertz contact solution. Where, the machine tool settings are the basic unknown variables for both design and manufacturing. This method integrating tooth flank design with manufacturing can accelerate the development efficiency of aerospace spiral bevel gear product [40].
- iv) With application of the proposed MOO machine tool settings modification, it can get a significant attempt to the recent collaborative manufacturing considering both geometric and physical performances [40]. It can extend the recent collaborative manufacturing for a case that higher loaded contact performances were simultaneously controlled and optimized within the qualified scopes for aerospace spiral bevel gears.”
- Discuss the limitations of the proposed method.
RES: In conclusions part, we highlight the shortcoming of the work, as follows:
“In this work, a data-driven MOO requiring loaded meshing transmission performances is proposed for aerospace spiral bevel gears by systematically modifying the machine tool settings. There are several distinct features as follows:
- i) Distinguished from the traditional SLTCA method using economical finite element software [38], the proposed NLTCA is reliable and time-saving for optimization of loaded meshing transmission performances. Moreover, this numerical method can provide some accesses to collaborative optimization considering both geometric performances and physical performances.
- ii) Data-driven accurate model of the loaded meshing transmission performance MOO in collaborative consideration of the loaded contact pressure distribution, contact elastic deformation, loaded contact pattern, and loaded transmission error is provided. Its nature allows the gear designer to get a good practicability for the future advanced gear design by setting the corresponding objective functions. Moreover, the tooth flank contact fatigue performance relating to residual stress [58,59], micro surface topography [60] and surface roughness [61] should be taken into accounts in the proposed MOO and this design can get an important access to high-performance tooth flank manufacturing of aerospace spiral bevel gears.
iii) The proposed method can get a basic input for the future high-performance design considering complex operation conditions such as the high speed, coupling effect of the multiple filed and lubrication. It means that the loaded meshing transmission performance can be optimized within the prescribed scopes by fine-modifying machine tool settings. Moreover, considering the various manufacturing errors, the experimental verification for MOO results by improving the employed algorithm [62, 63] would be the main task in future work.”
- The conclusion section needs to be redrawn.
RES: In this revision, we provide a corresponding correction, as follows:
“In this work, a data-driven MOO requiring loaded meshing transmission performances is proposed for aerospace spiral bevel gears by systematically modifying the machine tool settings. There are several distinct features as follows:
- i) Distinguished from the traditional SLTCA method using economical finite element software [38], the proposed NLTCA is reliable and time-saving for optimization of loaded meshing transmission performances. Moreover, this numerical method can provide some accesses to collaborative optimization considering both geometric performances and physical performances.
- ii) Data-driven accurate model of the loaded meshing transmission performance MOO in collaborative consideration of the loaded contact pressure distribution, contact elastic deformation, loaded contact pattern, and loaded transmission error is provided. Its nature allows the gear designer to get a good practicability for the future advanced gear design by setting the corresponding objective functions. Moreover, the tooth flank contact fatigue performance relating to residual stress [58,59], micro surface topography [60] and surface roughness [61] should be taken into accounts in the proposed MOO and this design can get an important access to high-performance tooth flank manufacturing of aerospace spiral bevel gears.
iii) The proposed method can get a basic input for the future high-performance design considering complex operation conditions such as the high speed, coupling effect of the multiple filed and lubrication. It means that the loaded meshing transmission performance can be optimized within the prescribed scopes by fine-modifying machine tool settings. Moreover, considering the various manufacturing errors, the experimental verification for MOO results by improving the employed algorithm [62, 63] would be the main task in future work.”
Reviewer 4 Report
1Comments:.
1) Need to mention & discuss the type of load that is applied for the specified gears in the introduction section.
2 2) Need to mention about the type of meshing and their mesh size used in this research.
3 3) The authors can present more details on the process of material selection which may help the readers in application context.
4) Do the authors consider wear rate in loaded condition of meshing and transmission.
5 5) Authors can include the ANSYS simulation and analysis images in the paper.
6 6) Line 576, 577, check the sentence formation.
Author Response
- Need to mention & discuss the type of load that is applied for the specified gears in the introduction section.
RES: indeed, we pay an attention on this point.
In current field of spiral bevel gears, we focused the setup about the applied load, as follows:
“The applied torque is 250N-M for gear axis. In consideration of the computational efficiency, the boundary condition is that shell bottom is clamped along the circular edge and the tooth tip, concave, convex, heel and toe are free. For the design data and design procedure, the detailed determination can refer to the Ref.[43].”
- Need to mention about the type of meshing and their mesh size used in this research.
RES: In this revision, we provide the detailed explanations, as follows:
“With the given basic design data on gear blank, head-cutter and hypoid generator, data-driven tooth flank modeling is performed. Fig.6 shows three-dimensional solid model based on computer-aided design (CAD) software and the finite element model based on software ABAQUS [43]. To get accurate calculation results, the tooth profiles are meshed with solid element, and the type of element is the hexahedron reduced integral element C3D8R. The material properties of the finite element model is set the same with the given material parameters [44] that Young’s modulus is 2.09´105 MPa and the Poisson’s ratio is 0.3. As for finite element model of aerospace spiral bevel gear, the total number of finite elements is 360656 and the number of nodes is 430576.”
- The authors can present more details on the process of material selection which may help the readers in application context.
RES: In current field of spiral bevel gears, we generally ignored the material.
4) Do the authors consider wear rate in loaded condition of meshing and transmission.
RES: Wear is very complex problem. Actually, the method proposed in this paper is still far from the real wear.
5) Authors can include the ANSYS simulation and analysis images in the paper.
RES: Actually, ANSYS application is always similar with the employed ABAQUS in this work, as follows:
“With the given basic design data on gear blank, head-cutter and hypoid generator, data-driven tooth flank modeling is performed. Fig.6 shows three-dimensional solid model based on computer-aided design (CAD) software and the finite element model based on software ABAQUS [43]. To get accurate calculation results, the tooth profiles are meshed with solid element, and the type of element is the hexahedron reduced integral element C3D8R. The material properties of the finite element model is set the same with the given material parameters [44] that Young’s modulus is 2.09´105 MPa and the Poisson’s ratio is 0.3. As for finite element model of aerospace spiral bevel gear, the total number of finite elements is 360656 and the number of nodes is 430576.”
6) Line 576, 577, check the sentence formation.
RES: In this revision, we made a corresponding modification.
Round 2
Reviewer 1 Report
There has been a clear improvement in the presentation, and most of the questions have been accessed.
I have no further request, and, the article is now in a acceptable in this present form.
Reviewer 4 Report
Reviewers comments were suitably addressed and therefore, the manuscript may be accepted